# SocialVeil: Probing Social Intelligence of Language Agents under Communication Barriers

## Abstract

Large language models (LLMs) are increasingly evaluated in interactive environments to test their social intelligence. However, existing benchmarks usually assume idealized communication between agents, limiting our ability to diagnose whether LLMs can maintain and repair interactions under real-world conditions. To close this gap, we present SocialVeil, a social learning environment that can simulate social interaction under cognitive-difference-induced communication barriers. SocialVeil introduces three representative types of such disruption, *semantic vagueness*, *sociocultural mismatch*, and *emotional interference*. SocialVeil also introduces barrier-aware evaluation metrics, *unresolved confusion* and *mutual understanding*, which complement standard goal-oriented evaluation by assessing agents' capability of maintaining interaction in impaired communication. Experiments across 720 scenarios and four frontier LLMs show that barriers consistently impair performance, with mutual understanding reduced by over 45% on average, and confusion elevated by nearly 50%. Human evaluations validate the fidelity of these simulated barriers (ICC≈0.78, Pearson r≈0.80). We further demonstrate that adaptation strategies (Repair Instruction and Interactive learning) only have a modest effect that remains far from barrier-free performance. This work takes a step toward bringing social interaction environments closer to real-world communication, opening broader opportunities for exploring the social intelligence of LLM agents.

## 1 Introduction

> *"Communication is not the transmission of a message, but the negotiation of meaning." – Karl Weick*

Human social interaction is inherently dynamic and suffused with uncertainty. Conversations are not mere exchanges of facts but complex processes in which people navigate ambiguity, negotiate relational concerns, and repair misunderstandings (e.g., Clark, 1996). For example, between friends, people may deliberately phrase an idea vaguely, assuming that the intended meaning can be inferred from shared contextual knowledge (e.g., Grice, 1990). In professional settings, colleagues often resort to indirect refusals, strategically mitigating the threat to social harmony posed by outright disagreement (e.g., Brown & Levinson, 1987). Despite such uncertainties, conversations generally progress through interlocutors' flexible use of clarification, accommodation, and empathic engagement, which jointly sustain the coherence of interaction and the stability of social relationships.

We define these systematic factors that hinder mutual understanding in dialogue as **communication barriers**. Unlike incidental noise, such barriers are structured, patterned, and consequential, shaping how interlocutors interpret one another's words and craft their responses (e.g., Clark, 1996; Schegloff et al., 1977). Recognizing the role of communication barriers in shaping agent behavior is essential for developing socially-aware AI systems, particularly those designed to function effectively in complex and dynamic environments, as they expose subtle failure modes that aggregate metrics overlook and offer valuable diagnostic insight for more robust and responsible deployment.

However, constructing a principled framework for simulating communication barriers is challenging due to several aspects: 1) **Intractable Taxonomy.** Barriers manifest at many levels, from perceptual-level acoustic interference (e.g., Cherry, 1953) to discourse-level breakdowns (e.g., Schegloff et al.,

**Figure 1: Existing social interaction benchmark usually provides an idealized setting for social interaction.** Comparison between existing social interaction benchmarks (clean, idealized interactions) and real-world conversations (messy, ambiguous interactions), highlighting sociocultural mismatch, emotional interference, semantic vagueness, and mutual understanding challenges.

1977). Existing research still lacks a well-structured, literature-supported taxonomy to reliably guide systematic investigation. 2) **Realism–Control Tradeoff.** Barriers must remain faithful to social practice yet also be instantiated in a controlled and reproducible manner. Naive noise injection often breaks realism, whereas free-form prompts tend to sacrifice consistency and comparability. 3) **Metric Insufficiency.** The presence of a barrier does not always entail failure of social interaction; an agent may still accomplish its goal through brute-force strategies while undermining the relationship or mutual understanding. Thus, a barrier-aware environment must evaluate not only task success but also incorporate metrics that capture the broader effects of barriers.

Recent studies have introduced interactive environments and benchmarks to assess agents' social intelligence capability (e.g., Chen et al., 2024; Mou et al., 2024; Zhou et al., 2023). These works are typically constructed under idealized conditions. Agents are presumed to share the same linguistic assumptions, sociocultural norms, and emotional registers, thereby overlooking the vagueness, misalignment, and disruptions that pervade real interaction. As previously discussed and as illustrated in Figure 1, actual conversational settings are far less idealized, where communication barriers often give rise to misunderstandings or conflicts, creating a more complex interactional environment.

To emulate real-world interaction settings, we introduce SOCIALVEIL, a framework for creating a realistic and interactive social environment that evaluates agents' social intelligence capability under communication barriers. Through a systematic literature review, we identify three communication barriers rooted in cognitive factors: *Semantic Vagueness*, *Sociocultural Mismatch*, and *Emotional Interference*. We construct 180 episodes for each of the three barrier types as well as for a barrier-free baseline, resulting in a total of four sets of episodes. These scenarios are adapted from SOTOPIA (Zhou et al., 2023). To capture communication barriers' impact beyond social interaction task completion, we introduce a barrier-aware evaluation protocol and conduct a comprehensive evaluation of four frontier LLMs and verify their validity through comprehensive human evaluation. We further implement targeted interventions, such as repair-oriented instruction and the interactive learning framework, to enhance the agent's ability to engage in social interactions in barrier scenarios.

**Main Discoveries.** Our analysis demonstrates that communication barriers consistently impair agents' social intelligence capabilities. For example, *Semantic Vagueness* often prevents agents from establishing shared context, leading to substantial declines in mutual understanding (58% drop), and *Emotional Interference* often disrupts relationship quality (49% drop). We further validate the fidelity of simulated barriers and the barrier-aware evaluation protocol through human evaluation: annotators have shown strong inter-rater reliability (*avg ICC $\approx$ 0.78*), can successfully identify barrier types (*avg Accuracy $\approx$ 68%*), and show strong alignment with automatic metrics (*avg Pearson's r $\approx$ 0.80*). Additionally, we find that instruction-level interventions have only a minimal impact on agents' performance. In contrast, interactive learning (BC+SR) yields modest but consistent gains. Yet, these improvements remain insufficient to fully recover barrier-free performance, emphasizing the difficulty of achieving human-level social skills in the presence of communication barriers.

Overall, our contributions are as follows: 1) We introduce SOCIALVEIL, a barrier-aware, socially interactive environment for simulating and evaluating LLM agents' social intelligence under barrier

scenarios. 2) We propose an automated, barrier-aware evaluation protocol that complements conventional goal-oriented measures by explicitly capturing whether agents can maintain interaction and repair misunderstandings under communication barriers. 3) We demonstrate that barriers simulated in SOCIALVEIL induce effects that align with their real-world counterparts, validating the framework's fidelity as a proxy for studying real-world interaction. 4) We explore adaptation strategies for enhancing agents' performance under barriers, showing that while instruction-level interventions are largely ineffective, interactive learning yields steady yet limited improvements—highlighting both the promise of adaptive training and the remaining gap to human-level resilience.

## 2 SOCIALVEIL: A BARRIER-AWARE SOCIAL ENVIRONMENT

To better reflect real-world conditions, we propose a social learning environment with the following desiderata: 1) **Task Agnostic.** Barriers should keep social goals and context intact. 2) **Structured Disruptions.** Barriers should be systematically designed to have an intended level of disruption. 3) **Barrier-aware Evaluation.** Environment must support evaluation that moves beyond goal-oriented dimensions to capture the nuanced social and relational failures induced by the barriers. Motivated by these goals, we introduce SOCIALVEIL, a social learning environment that systematically embeds realistic communication barriers into simulated interactions. An overview is shown in Figure 2.

### 2.1 BARRIER TAXONOMY

In this study, we focus primarily on barriers that affect understanding, reasoning, and decision-making (stemming from cognitive factors), rather than on external, physical barriers that hinder the transmission of information (such as loud noise). Through a systematic review of research on interaction and communication, we identified three major categories of communication barriers induced by cognitive factors: *Semantic Vagueness*, ambiguity from vague pronouns or placeholders; *Socio-cultural Mismatch*, misaligned interpretations across cultural communication styles; and *Emotional Interference*, affective intensity obscuring task content. Table 1 provides definitions, examples, and theoretical grounding for each barrier (details of the literature review presented in Appendix B).

| Barrier Type | Definition | Real-world Example | Theoretical Grounding |
|---|---|---|---|
| **Semantic Vagueness** | Explicit referents are substituted with indeterminate pronouns or empty placeholders, leaving interpretation underspecified and prone to ambiguity. | *"It might work... you know what I mean."* | Pragmatics (Grice, 1990); Hedges (Lakoff, 1973); Fuzzy logic (Zadeh, 1965) |
| **Sociocultural Mismatch** | Cultural differences in communication styles lead to misaligned interpretations and hinder explicit understanding. | *"We'll think about it." Taken as postponement, but meant as refusal.* | Politeness (Brown & Levinson, 1987); Context theory (Hall, 1976); Linguistic relativity (Sapir, 1929) |
| **Emotional Interference** | Affective intensity overrides informational clarity, displacing task-relevant content with expressive overflow. | *"I'm too upset to explain—just figure it out yourself!"* | Attention (Eysenck et al., 2007); Emotion regulation (Gross, 1998b); Appraisal (Lerner & Keltner, 2000); |

**Table 1: Three types of communication barriers in SOCIALVEIL.** They are theoretically grounded, aligned with real-world interaction patterns, and operationalized in our environment.

### 2.2 BARRIER DESIGN

In SOCIALVEIL, we instantiate communication barriers unilaterally. One agent, designated as the barrier agent, communicates under a chosen barrier condition, while the other, the partner agent, remains in standard settings. Each barrier $b \in B$ is instantiated by composing a style prompt $P_b$ with a parameterization $R_b$ over four operational dimensions: **Narrative Stance** (*global pragmatic prior, e.g., indirectness/affect priority*), **Interaction Tactics** (*surface devices such as hedging, shell nouns, indirect refusals*), **Confusion Mechanisms** (*procedural constraints that resist disambiguation, e.g., withhold confirmation/deflect clarification*), and **Exemplar Templates** (*canonical utterance patterns for calibration/reproducibility; non-prescriptive*).

In practice, each barrier $b$ is implemented through a two-layer design. The style prompt $P_b$ encodes a high-level directive injected only into the barrier agent (*e.g., "overuse pronouns and ellipses" for semantic vagueness*), while the parameterization $R_b$ specifies quantitative cues that render the behavior reproducible. At runtime, $P_b$ and $R_b$ are combined only for the barrier agent, whereas

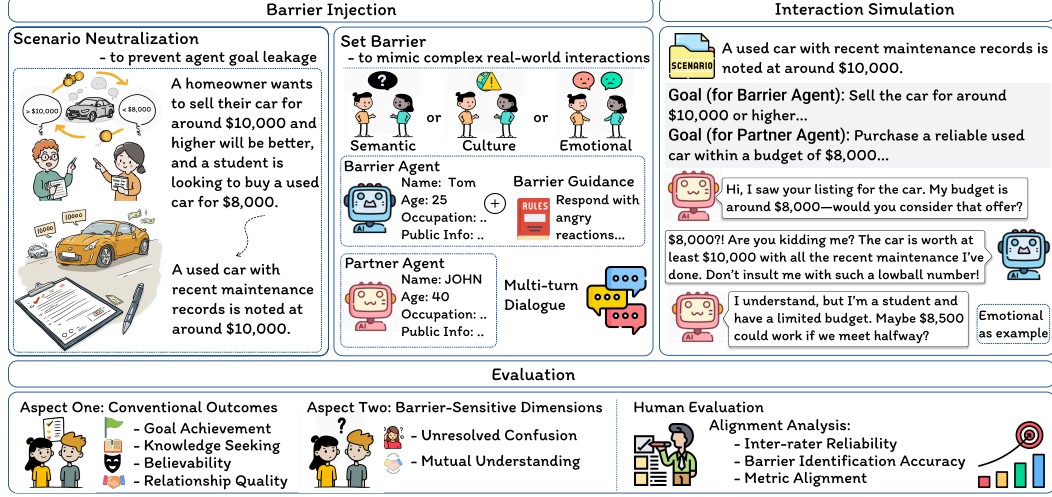

**Figure 2: Overview of SOCIALVEIL.** The pipeline consists of three stages: (**Barrier Injection**) scenarios are neutralized to remove bias and barriers are systematically injected; (**Interaction Simulation**) agents engage in multi-turn dialogue under barrier conditions; (**Evaluation**) agent performance is assessed using automatic metrics (goal achievement, knowledge seeking, believability, relationship quality, confusion, mutual understanding) and human evaluation (alignment analysis and barrier navigation success).

the partner agent remains unmodified ($P_b = \emptyset$, $R_b = \emptyset$), ensuring the barrier is the sole source of disruption and making evaluation controlled and repeatable.

## 2.3 SIMULATION SETUP

**Episode Design.** A SOCIALVEIL episode is a two-agent role-play, where each agent is assigned a private social goal and role profile. One agent, designated as the barrier agent, communicates under a chosen barrier condition, while the other, the partner agent, remains unmodified. This asymmetric design reflects natural human scenarios: for instance, a colleague whose indirect style obscures intent, or a teammate whose emotions color their contributions. To construct episodes, we adapt scenarios from existing social benchmarks and apply a neutralization step to their public scenario descriptions, which may otherwise leak the agent's private goals. Specifically, an LLM (*e.g.*, GPT-4o) rewrites each scenario description following fixed instructions to preserve contextual setting and role consistency while eliminating goal-related hints. This ensures that both agents share the same public background information but cannot trivially infer each other's private goals.

Formally, an episode is defined as

$$E = (\mathcal{A}_b, \mathcal{A}_p, g_b, g_p, p_p, p_b, b),$$

where $\mathcal{A}_b$ and $\mathcal{A}_p$ denote the barrier and partner agents, $g_b$ and $g_p$ their respective goals, $p_b$ and $p_p$ their role profiles, and $b$ the injected barrier type.

**Utterance Generation.** At each dialogue turn $t$, let $h_t$ denote the history of all utterances prior to $t$. Each agent $\mathcal{A}i$ generates an utterance $u_{t,i}$ conditioned on $h_t$, its goal $g_i$, and profile $p_i$. For the barrier agent $\mathcal{A}_b$, the instruction $I$ is augmented with the barrier specification $b$:

$$u_{t,b} \sim \pi_\theta(\cdot \mid h_t, g_b, p_b, I \oplus b),$$

The partner agent $\mathcal{A}_p$, by contrast, generates from the unmodified instruction:

$$u_{t,p} \sim \pi_\theta(\cdot \mid h_t, g_p, p_p, I),$$

This unilateral setup reflects natural conversational asymmetry, where communication difficulties typically arise from one interlocutor. It also allows us to evaluate the partner agent's performance when faced with impaired communication.

## 2.4 EVALUATION PROTOCOL

A barrier-aware social environment must judge not only task success but also whether agents remain socially competent under impaired communication. SOCIALVEIL, therefore, evaluates social interaction along two complementary aspects: The first aspect focuses on goal-oriented dimensions such as goal completion, relationship quality, and knowledge, which are widely measured in previous social interaction research. The second aspect introduces barrier-aware dimensions, which directly target the communicative disruptions induced by barriers. Specifically, we introduce 1) *Unresolved Confusion* to quantify the extent to which ambiguity remains at the end of the dialogue (five-point Likert scale, from incoherent to fully resolved), and 2) *Mutual Understanding* to capture the degree of convergence on shared context and goals (five-point Likert scale, from complete misalignment to full alignment). The details of the evaluation protocol implementation are shown in Appendix E

## 3 RESEARCH QUESTIONS AND EXPERIMENT SETUP

Our aim is not only to design systematic communication barriers within social interaction scenarios, but also to investigate how barriers influence agent performance. We frame our study around three guiding questions: 1) **Barrier Validity.** Do injected barriers reliably create structured disruptions? 2) **Barrier Effects.** How do different types of barriers affect the performance of frontier LLMs in social interaction? 3) **Barrier Adaptation.** Can agents be improved to handle communication barriers? To answer these questions, we build a testbed of 180 episodes for each barrier type (*Semantic Vagueness*, *Sociocultural Mismatch*, and *Emotional Interference*), along with a baseline condition without barriers, following the procedure described in Sec. 2. All episodes are drawn from SOTOPIA scenarios (Zhou et al., 2023), and we employ GPT-4o-mini as the base model for the barrier agent.

To investigate barrier effects, we evaluate four partner agents representing both proprietary and open-weight model families: GPT-4o-mini (Hurst et al., 2024), Qwen2.5-7B-Instruct, Qwen3-4B-Instruct (Yang et al., 2025), and Mistral-8B-Instruct (AI, 2024). The agents' generation temperature is set to 0.7 to encourage response diversity. Performance is measured using the automatic evaluation protocol described in Sec. 2.4, where we use GPT-4o as the backbone of the evaluator model, with its temperature set to 0.0 for stable, deterministic judgments.

To investigate whether agents can be made more resilient to communication barriers, we implement two strategic adaptations: 1) **Repair Instruction.** We first examine a direct, instruction-based intervention. This approach enhances the partner agent's meta-prompt with explicit guidance designed to reduce misunderstandings—for example, *"Actively ask clarifying questions and paraphrase to confirm understanding."* 2) **Interactive Learning.** We adapt an interactive learning framework (e.g., Wang et al., 2024). The process begins with behavior cloning (BC), where expert trajectories are generated from interactions using GPT-4o as partner agents and filtered for success using our evaluation protocol. The partner agent is initialized by imitating these trajectories through training:

$$\mathcal{L}(\theta) = -\mathbb{E}_{(h_t, u_t^*) \sim \mathcal{D}} \left[ \log \pi_\theta(u_t^* \mid h_t) \right], \tag{1}$$

where $\mathcal{D}$ is the set of high-quality demonstrations and $u_t^*$ is the expert utterance given history $h_t$. Then, we apply self-reinforcement (SR), the trained agent engages with the fixed barrier agent to produce new dialogues, from which high-quality trajectories are again filtered and added to training. This iterative process allows the agent to progressively distill strategies for navigating barriers.

## 4 EXPERIMENT RESULTS

### 4.1 ARE THE CREATED BARRIER VALID?

In sociolinguistics and communication theory, barriers are viewed not as random noise but as structured phenomena that systematically distort interactional signals (Clark & Brennan, 1991; Tannen, 2005). To claim validity, a simulated barrier should therefore induce structured distribution shifts in how communication is represented, rather than producing arbitrary variation.

We tested this by probing the hidden states of Qwen2.5-7B-Instruct in the baseline and three barrier conditions. As shown in Figure 3, barrier conditions form distinct and compact clusters in the t-SNE space, with a clear separation from baseline points. This indicates that barriers are encoded in the model's internal representations as structured modes of variation, rather than as random noise.

| Model | Barrier Type | SOTOPIA-all | | | | | | SOTOPIA-hard | | | | | |
|---|---|---|---|---|---|---|---|---|---|---|---|---|---|
| | | BEL | REL[†] | KNO | GOAL | Confus.[‡] | Mutual[§] | BEL | REL[†] | KNO | GOAL | Confus.[‡] | Mutual[§] |
| GPT-4o-mini | Baseline | 8.78 | **3.41** | 3.94 | 7.60 | 4.08 | **4.56** | 8.82 | **2.53** | 2.78 | 6.75 | 3.75 | **4.46** |
| | Semantic | 7.58 | 1.91 | 2.89 | 5.61 | 1.48 | 1.76 | 7.32 | 1.61 | 2.71 | 5.46 | 1.57 | 1.78 |
| | Socioculture | 7.70 | 1.95 | 2.92 | 5.35 | 1.78 | 2.41 | 7.61 | 1.64 | 2.96 | 5.32 | 1.75 | 2.28 |
| | Emotional | 7.76 | 1.61 | 2.97 | 5.25 | 1.51 | 2.03 | 7.42 | 1.03 | 2.93 | 4.96 | 1.46 | 1.89 |
| Qwen2.5-7b-Instruct | Baseline | 8.48 | **3.17** | 3.79 | 7.48 | 4.06 | **4.45** | 8.33 | **1.72** | 2.73 | 5.68 | 3.16 | **3.89** |
| | Semantic | 7.37 | 1.91 | 2.63 | 5.99 | 1.61 | 1.91 | 6.96 | 1.46 | 2.25 | 5.25 | 1.21 | 1.39 |
| | Socioculture | 7.65 | 2.04 | 2.79 | 5.71 | 1.85 | 2.45 | 7.28 | 1.64 | 2.61 | 5.17 | 1.57 | 1.93 |
| | Emotional | 7.56 | 1.57 | 2.64 | 5.47 | 1.63 | 2.16 | 7.28 | **0.89** | 2.28 | 4.75 | 1.39 | 1.78 |
| Qwen3-4b-Instruct | Baseline | 8.64 | **3.12** | 3.88 | 7.73 | 3.72 | **4.30** | 8.40 | **1.96** | 2.93 | 6.57 | 3.14 | **4.03** |
| | Semantic | 7.80 | 2.03 | 2.85 | 6.81 | 1.97 | 2.42 | 7.61 | 1.32 | 2.28 | 6.04 | 1.68 | 1.86 |
| | Socioculture | 7.89 | 2.04 | 3.10 | 6.45 | 2.26 | 3.02 | 7.64 | 1.28 | 3.03 | 5.96 | 2.03 | 2.64 |
| | Emotional | 7.94 | 1.72 | 3.06 | 6.48 | 1.94 | 2.64 | 7.75 | 1.21 | 2.75 | 5.78 | 1.67 | 2.32 |
| Mistral-8b-Instruct | Baseline | 7.73 | **2.84** | 3.74 | 6.83 | 3.23 | **3.54** | 7.73 | **2.42** | 3.39 | 6.22 | 2.61 | **3.25** |
| | Semantic | 7.01 | 1.56 | 2.41 | 3.91 | 1.07 | 1.13 | 6.81 | 1.32 | 2.25 | 3.50 | 1.01 | **1.00** |
| | Socioculture | 7.81 | 2.28 | 3.02 | 5.31 | 1.52 | 1.85 | 6.45 | 2.07 | 3.07 | 5.07 | **1.21** | 1.43 |
| | Emotional | 7.47 | 1.39 | 2.69 | 4.48 | 1.26 | 1.42 | 6.48 | 1.28 | 2.79 | 4.03 | 1.11 | 1.18 |

**Table 2: Barriers consistently degrade agent performance compared to the baseline across all models.** All results use GPT-4o-mini to act as a barrier agent. Semantic vagueness causes the sharpest drop in clarity, emotional interference disproportionately harms relationships, and sociocultural mismatch induces persistent confusion.

## 4.2 HOW DO BARRIERS AFFECT SOCIAL INTERACTION?

Table 2 summarizes each model's performance across baseline and barrier conditions. From these results, we distill three key findings: 1) **Barriers consistently impair social interaction performance.** Across all models and evaluation dimensions, the presence of barriers leads to significant performance degradation compared to the baseline. This effect holds for metrics from both goal-oriented dimensions and barrier-aware dimensions. 2) **Barrier types exhibit distinct patterns.** Each barrier produces a characteristic pattern of degradation. Semantic vagueness most severely disrupts mutual understanding ($avg$ $-58\%$), often preventing agents from converging on shared context. Emotional interference disproportionately damages the quality of the relationship ($avg$ $-49\%$), while sociocultural mismatch induces persistent confusion ($avg$ $-49\%$) with relatively mild relational effects. 3) **Social reasoning is more fragile than goal pursuit.** Compared to goal completion and knowledge acquisition, which decline

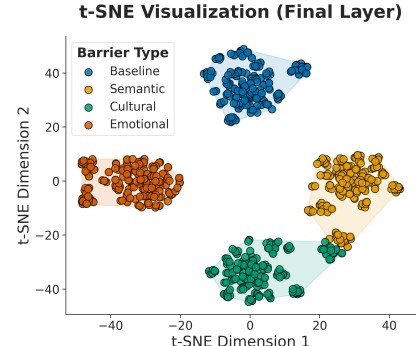

**Figure 3: t-SNE visualization of the final-layer representations from Qwen2.5-7B-Instruct**, showing clear clustering by barrier type. Each cluster forms a distinct region in the embedding space, highlighting the separability of barrier categories.

moderately under barriers ($20 - 30\%$), social dimensions suffer substantially greater degradation, with relationship quality dropping by an average of $45\%$ and mutual understanding declining by $52\%$ across all barrier types, which infers barriers primarily disrupt the subtle social reasoning required.

## 4.3 CAN AGENTS ADAPT TO COMMUNICATION BARRIERS?

Table 3 reports the comparison of agent social interaction performance under communication barriers between the default setting and the two adaptation strategies. We can draw several key findings: 1) **Repair Instruction yields trivial performance improvements.** This result highlights two points. First, overcoming communication barriers is not a trivial skill that can be invoked by an instruction-level guidance; it requires agents to detect when breakdowns occur, attribute them to a specific distortion, and deploy targeted repair strategies. Second, the failure of this strategy reflects the limitation of static, barrier-agnostic prompting: agents often resort to shallow repetitions

---

[2]Relationship Quality is most impacted by the *Emotional* barrier (e.g., **0.89**).

[3]Confusion is most prominent under the *Sociocultural* barrier (e.g., **1.21**).

[4]Mutual Understanding is most degraded by the *Semantic* barrier (e.g., **1.00**).

or generic clarifications that do not resolve the underlying disruption. 2) **Interactive Learning (BC+SR) yields consistent but limited improvements.** In contrast, interactive learning produces steady gains across all barrier types and relieves agents' struggle with social interaction in barrier cases. Yet the improvements remain modest (*avg 10-20%*), such that the enhancement over Repair Instruction is incremental rather than dramatic, and performance still falls significantly short of the barrier-free baseline.

| Model | Barrier | GOAL | | | MUTUAL | | | CONFUSION | | |
|-------|---------|------|--------|--------|------|--------|--------|------|--------|--------|
| | | Base | Repair | (BC+SR) | Base | Repair | (BC+SR) | Base | Repair | (BC+SR) |
| | Semantic | 5.99 | 6.07 | 6.02 | 1.91 | 1.90 | **2.15** | 1.61 | 1.66 | **1.84** |
| Qwen2.5-7B | Sociocultural | 5.71 | 5.87 | 5.79 | 2.45 | 2.50 | **2.60** | 1.85 | 1.88 | **2.16** |
| | Emotional | 5.47 | 5.74 | **5.86** | 2.16 | 2.28 | **2.34** | 1.63 | 1.69 | **1.85** |
| | Semantic | 6.81 | 7.08 | **7.13** | 1.97 | 1.98 | **2.10** | 2.42 | 2.48 | **2.49** |
| Qwen3-4B | Sociocultural | 6.45 | 6.85 | 6.84 | 2.26 | 2.43 | **2.49** | 3.02 | 3.08 | **3.21** |
| | Emotional | 6.48 | 6.55 | 6.52 | 1.94 | 1.92 | **2.09** | 2.64 | 2.39 | 2.46 |

**Table 3. Repair instructions are ineffective, while interactive learning offers modest gains.** Repair instruction yields trivial improvements ($< 2\%$). Interactive learning (BC+SR) improves mutual understanding by 8.6% on average, though confusion increases by 7.8%, and still falls short of barrier-free levels.

3) **Both Strategies show minimal impact on goal completion.** Notably, neither approach benefits the GOAL scores compared to the baseline, suggesting that these adaptation strategies may guide agents to focus primarily on managing communication barriers rather than achieving social objectives. This indicates a potential trade-off where barrier-handling mechanisms divert cognitive resources away from goal-oriented behavior, highlighting the challenge of simultaneously maintaining task performance while addressing communication disruptions.

## 5 DISCUSSION

### 5.1 ANALYSIS: BEHAVIORAL ALIGNMENT OF SIMULATED BARRIERS

To further analyze the effects of injected barriers, we move beyond metrics to a behavioral analysis of the conversation trajectory. Inspired by the previous work on evaluating the alignment of simulated behaviors (Park et al., 2023; Han et al., 2025), we design a series of qualitative analyses to study the alignment between our simulated barriers and their real-world counterpart.

**Linguistic Signatures.** We first test whether barriers trigger systematic linguistic shifts consistent with real-world communication breakdowns. Empirically, we extracted four linguistic features from conversations: reference pronouns (e.g., "it," "that") (Ariel, 2014), hedging words (e.g., "maybe," "could") (Lakoff, 1973), sentiment polarity, and self-focus pronouns (e.g., "I," "my") (Pennebaker, 2011) and correlated them with metrics in the evaluation protocol.

Figure 4 reveals two patterns: 1) Reference pronouns and self-focus are negatively associated with conversational quality, where they show correlation with higher confusion and lower mutual understanding; and 2) Sentiment polarity serves as a positive predictor of smooth social interaction, where a more positive tone aligns with better relationships and greater goal attainment.

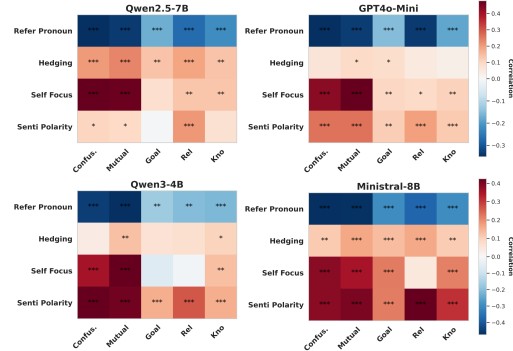

**Barrier-specific Effects.** We quantify each barrier's unique effect as its deviation from the mean of the other two barriers on the same metric, model, and scenario, with bootstrap 95% confidence intervals (Figure 5). The results further support that the effect of our simulated barriers

**Figure 4:** Linguistic signal & metrics correlations.

aligns with their real-world counterparts: For example, semantic barriers most strongly impair Mutual Understanding, while emotional barriers disproportionately erode Relationship Quality, and Cultural barriers uniquely elevate Unresolved Confusion. These effects are consistent across models and statistically significant in the expected directions, demonstrating that the simulated barriers

not only alter surface linguistic features but also reproduce distinct interactional disruptions characteristic of their real-world analogs.

## 5.2 ANALYSIS: HUMAN EVALUATION

To complement our automatic metrics, we conduct a human evaluation to validate that the created barriers manifest in ways consistent with their intended real-world counterparts. The technical details of the human evaluation process are available in Appendix D. We assess three aspects: 1) **Inter-rater Reliability.** Whether the human annotators are reliable, 2) **Barrier Identification Accuracy.** Whether the injected barriers can be identified by human annotators, and 3) **Metric Alignment.** Whether model-rated barrier-sensitive dimensions align with human judgments.

**Inter-rater Reliability.** We calculated Fleiss's Kappa for human annotation of barrier types, yielding a score of 0.38. Based on the classic interpretive framework of Landis & Koch (1977), this reflects fair agreement, bordering on moderate. Such values are common in complex, subjective multi-class annotation tasks. Prior studies report similar $\kappa$ ranges—for instance, Castro et al. (2019) reported values ranging from 0.23 to 0.59 for sarcasm detection, and Callejas & López-Cózar (2008) reported 0.32–0.42 for non-acted emotion identification. Thus, our result aligns well with similar topic findings, further supporting the overall reliability and consistency of our manual annotations.

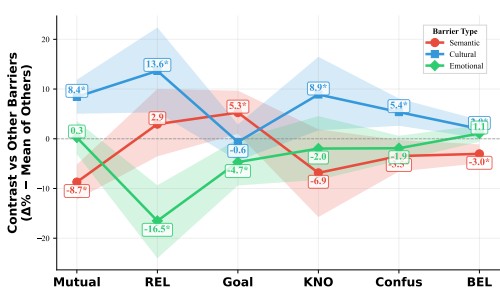

**Figure 5:** Barrier effects (mean ±95% CI).

For the two rating metrics, each scenario was rated by three of six annotators. We used a one-way random effects model with single-measure ICC(1,k) to assess inter-rater reliability. *Unresolved Confusion* showed ICC = 0.77, 95% CI [0.68, 0.83], $F(119, 240) = 4.26$, $p < .001$. *Mutual Understanding* showed ICC = 0.79, 95% CI [0.72, 0.85], $F(119, 240) = 4.80$, $p < .001$. Per Cicchetti (1994), these values fall in the good range, suggesting reliable inter-rater consistency.

**Barrier Identification Accuracy.** We further reported the accuracy of human annotators in identifying communication barriers. Given that the sample contained only 120 scenarios, we employed cluster bootstrap resampling with replacement at the scenario level, repeated 1000 times, to evaluate the stability of the estimates and to construct 95% percentile-based confidence intervals. From Figure 7, the results showed that the overall accuracy of human annotators in barrier identification was 0.68 (95% CI [0.63, 0.73]); with accuracy 0.76 (95% CI [0.67, 0.86]) for baseline barriers, 0.65 (95% CI [0.54, 0.76]) for semantic barriers, 0.63 (95% CI [0.53, 0.73]) for cultural barriers, and 0.67 (95% CI [0.54, 0.78]) for emotional barriers. These results demonstrate that, across all barrier types, annotator accuracy was significantly above the level expected from random guessing, as all lower confidence interval bounds exceeded the binary chance baseline of 0.50.

| Barrier | Human Evaluation | | | | Automated Evaluation | | | |
|---|---|---|---|---|---|---|---|---|
| | CONFUSION | | MUTUAL | | CONFUSION | | MUTUAL | |
| | Mean | 95% CI | Mean | 95% CI | Mean | 95% CI | Mean | 95% CI |
| Baseline | 3.74 | [3.48, 4.00] | 3.84 | [3.62, 4.07] | 3.94 | [3.65, 4.24] | 4.47 | [4.27, 4.67] |
| Semantic | 1.45 | [1.30, 1.62] | 1.71 | [1.56, 1.87] | 1.54 | [1.32, 1.79] | 1.81 | [1.52, 2.13] |
| Sociocultural | 1.94 | [1.67, 2.23] | 2.25 | [2.01, 2.51] | 1.67 | [1.50, 1.82] | 2.24 | [1.96, 2.50] |
| Emotional | 1.70 | [1.53, 1.86] | 1.96 | [1.80, 2.11] | 1.67 | [1.50, 1.84] | 2.20 | [1.92, 2.50] |

**Table 4:** Mean ratings by barrier type with 95% confidence intervals. Our results show strong alignment between human annotators and automated evaluation in rating *Mutual Understanding* and *Unresolved Confusion*. Pearson correlation: Confusion $r=0.80$ (95% CI [0.72, 0.86]), Mutual $r=0.79$ (95% CI [0.71, 0.85]). $n=120$.

**Metric Alignment.** We further compared the mean values and 95% confidence intervals of *Unresolved Confusion* and *Mutual Understanding* between human annotators and the model under different barrier conditions. As shown in Table 4, the model's performance on these two metrics was close to the average human ratings, with only small differences in the confidence intervals, indicating a high degree of consistency. In addition, the analysis of overall convergent validity revealed statis-

tically strong correlations between human and model scores, with *Unresolved Confusion* yielding a convergent validity of 0.80 (95% CI [0.72, 0.86]) and *Mutual Understanding* yielding a convergent validity of 0.79 (95% CI [0.71, 0.85]). These findings provide further evidence of the model's reliability in rating the barrier-sensitive dimensions.

# 6 RELATED WORKS

The dynamics of interactions among AI agents, as well as between AI and humans, have been examined across various disciplines. Our work builds upon existing research in social intelligence, its interactive assessment, agent-based social simulations, and the study of interaction and communication within the social sciences. For a more detailed discussion, see Appendix A.

## 6.1 STATIC BENCHMARKS FOR SOCIAL INTELLIGENCE AND THEIR LIMITATIONS

To assess the social intelligence of AI systems, researchers have proposed a wide range of static benchmarks. These draw inspiration both from clinical and psychological tests as well as from social commonsense reasoning tasks. For instance, ToMi is used to evaluate theory of mind in text comprehension (Le et al., 2019); FauxPas (social "faux pas" detection) examines models' ability to capture others' intentions and beliefs (Shapira et al., 2023b); SocialIQA focuses on event–intention–reaction commonsense question answering (Sap et al., 2019); and SocialIQ evaluates whether models can "read people" through multimodal video tasks (Zadeh et al., 2019). However, as model performance continues to improve, many of these datasets have reached near-saturation on certain subtasks, prompting the community to design more adversarial and challenging benchmarks (e.g., Shapira et al., 2023a). Still, existing research consistently highlights a fundamental limitation: static test items alone cannot capture the complexity and diversity of interactive settings. Consequently, there remains a significant gap in evaluating intelligence within real-world social interactions.

## 6.2 INTERACTIVE EVALUATION OF SOCIAL INTELLIGENCE AND AGENTS SIMULATION

LLMs encode rich knowledge and can generate human-like responses in social contexts (e.g., Park et al., 2023; West et al., 2021). Researchers have leveraged them for simulating social interactions, from optimizing social media design (Park et al., 2022) to building agents with credible human behavior (Park et al., 2023) and supporting collaborative software development (Qian et al., 2023). However, most existing studies primarily highlight the potential of LLMs in simulating social interactions, while systematic evaluation of agent performance in such interactions remains scarce. To address this gap, SOTOPIA (Zhou et al., 2023) builds on prior research in social simulation and dialogue systems to propose an open-ended interactive environment. This environment dynamically evaluates agents across multi-turn social scenarios, measuring their ability to achieve social goals and maintain role consistency, thereby overcoming the limitations of static benchmarks. Subsequent extensions of SOTOPIA advance interactive evaluation and learning. For example, SOTOPIA-$\pi$ introduces mechanisms for interactive imitation and reinforcement learning, enhancing agents' adaptability and strategic behavior (Wang et al., 2024); LIFELONG-SOTOPIA links events with memory to assess behavioral consistency in long-term interactions (Goel & Zhu, 2025); and SOTOPIA-RL incorporates fine-grained reinforcement signals for credit assignment and optimization at the discourse level (Yu et al., 2025). Taken together, these developments demonstrate the utility of SOTOPIA as a benchmark and a platform for advancing the social intelligence capabilities of AI agents.

# 7 CONCLUSION

In this work, we introduce SOCIALVEIL, a barrier-aware social interaction environment designed to simulate and evaluate the performance of LLM agents in the presence of communication barriers. Our experiments clearly show that these barriers consistently impair agents' social intelligence capabilities. Through human evaluations, we validate both the realism of the simulated barriers and the robustness of the evaluation protocol. We also investigate adaptation strategies, finding that repair instructions are largely ineffective, while interaction-driven learning yields modest but consistent improvements. This work represents a crucial step toward more realistic social interaction environments and opens promising new avenues for advancing the social intelligence of LLM agents.

ETHICS STATEMENT

This study investigates the social intelligence of language agents under communication barriers through controlled simulations. To validate the effectiveness and reliability of our automatic evaluation protocol, we recruited human participants from university subject pools. These participants contributed to an evaluation phase and received research credit for their involvement. All procedures involving human participants were reviewed in accordance with institutional ethical guidelines. Further details can be found in Appendix D. Our methods are used solely within a research context and are not deployed in real-world applications. Although our findings underscore the limitations of current models in nuanced communicative contexts, we aim to contribute to the development of safer, more responsible, and culturally inclusive AI systems. We believe that tools like SOCIALVEIL offer promising directions for mitigating sociotechnical risks in future language agent deployment.

REPRODUCIBILITY STATEMENT

We have implemented several steps to ensure the reproducibility of our work. Our methodology for barrier injection and interaction simulation is detailed in Sec. 2.1, Sec. 2.2, and Sec. 2.3. Evaluation protocols and barrier-aware metrics are described in Sec. 2.4, and additional implementation details (including prompts and hyperparameters) are provided in Appendix F. We will release our source code and data in camera ready version.

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

## A    EXTENDED RELATED WORKS

### A.1    FROM "SEAMLESS BY DEFAULT" TO "COGNITIVE BIAS INDUCED BARRIERS"

Despite recent progress in multi-agent evaluation and social agent simulation, many studies still operate under overly idealized assumptions. Early work often did not even differentiate between agents' access to information. Even in frameworks such as SOTOPIA and related research, where agents are designed to have little or no knowledge of one another's strategies or mental states, the communicative process itself is still frequently treated as seamless by default. In other words, once an utterance is produced, it is assumed to be correctly understood, with cooperative partners always willing to clarify and reach consensus. Such assumptions, however, diverge sharply from messy real-world interactive contexts. In actual human–machine and machine–machine communication, ambiguity, omission, misunderstanding, and cognitive bias are the norm. Communication is not a one-way transmission but a dynamic negotiation, shaped by obstacles, repair, and continuous adaptive feedback loops. Traditional dialogue and robotics research have long emphasized miscommunication detection and recovery, addressing challenges such as ambiguous instructions, misinterpretations arising from path or environmental constraints, and strategies for clarification based on situational evidence. This line of work underscores a fundamental truth: effective real-world communication does not mean *"never making errors,"* but rather possessing the capacity to handle errors once they occur. Importantly, such resilience and repair mechanisms are themselves central manifestations of social intelligence. Recent empirical surveys of multi-agent systems (Cemri et al., 2025) highlight that MAS do not always outperform single-agent systems across all dimensions. Many of these issues are closely tied to communicative misalignment, inconsistent role assumptions, and information loss both within and across interaction rounds. These findings call for moving beyond idealized models toward a systematic study of communication barriers and their diagnosability.

## B    BARRIER TAXONOMY DEVELOPMENT

In this study, we adopted the method of a systematic literature review to examine research in the social sciences on interaction and communication. Specifically, we focused on interaction/communication barriers caused by cognitive differences. Based on existing psychological and sociological theories related to interaction and communication, we conducted a literature search and categorized the findings. Ultimately, we identified three main types of communication barriers induced by cognitive differences: ***Semantic Vagueness***, ***Sociocultural Mismatch***, and ***Emotional Interference***.

### B.1    SEMANTIC VAGUENESS

Grice's theory of conversational implicature and the cooperative principle (Grice, 1990) clearly shows that communication fundamentally relies on shared maxims; when speakers intentionally or unintentionally violate the maxim of manner, interpretive ambiguity easily arises. Swinney's cross-modal lexical priming experiments (Swinney, 1979) compellingly demonstrated that multiple meanings of an ambiguous word are briefly activated in parallel, and without sufficient contextual cues, semantic comprehension quickly becomes confused. Frazier and Rayner (Frazier & Rayner, 1982) found that syntactic ambiguity can often trigger the garden-path effect, requiring substantial and effortful reanalysis by the listener. Zadeh's fuzzy set theory (Zadeh, 1965) and Lakoff's work on hedges (Lakoff, 1973) further illustrate how vague linguistic expressions can significantly expand or constrain conceptual boundaries, leading to diverse and sometimes divergent interpretations.

### B.2    SOCIOCULTURAL MISMATCH

The Sapir–Whorf hypothesis of linguistic relativity (e.g., Sapir, 1929; Whorf, 2012) emphasizes how habitual language use subtly shapes thought patterns, perception, and attentional focus. Hall's influential framework of high- versus low-context cultures (e.g., Hall, 1976; 1973) reveals cross-cultural variation in reliance on explicit verbal information versus shared, implicit contextual cues. Hofstede's cultural dimensions theory (e.g., Hofstede, 1984; 2001) highlights how variables such as power distance and individualism–collectivism influence communication styles, interpretive preferences, and behavioral expectations. Brown and Levinson's politeness theory (Brown & Levinson, 1987) and Giles' communication accommodation theory (e.g., Giles, 1973; Giles et al., 1991) demonstrate

how differing choices in politeness strategies and speech convergence or divergence across cultures can lead to misattributions of intent, communicative mismatches, and interpersonal friction.

### B.3 EMOTIONAL INTERFERENCE

Festinger's cognitive dissonance theory (e.g., Festiger, 1957; Festinger & Carlsmith, 1959) demonstrates how individuals engage in defensive cognitive processing when confronted with conflicting beliefs, attitudes, or information. Gross's influential emotion regulation model (e.g., Gross, 1998b;a) shows that suppression strategies often impair processing efficiency and can lead to heightened cognitive load. Lerner and Keltner's appraisal tendency framework (e.g., Lerner & Keltner, 2000) emphasizes the carry-over effect of emotions, whereby specific emotional states influence subsequent judgments, evaluations, and decisions beyond their original eliciting context. Eysenck's attentional control theory (e.g., Eysenck & Calvo, 1992; Eysenck et al., 2007) argues that anxiety weakens executive control functions, making individuals more susceptible to distraction by emotionally salient or threatening cues during communication. Slovic's affect heuristic (Gilovich et al., 2002) and Forgas's affect infusion model (Forgas, 1995) demonstrate how emotional states can directly shape both the interpretation of information and the strategies used in decision-making processes.

## C EXPERIMENTAL DETAILS

### C.1 ACRONYM FOR EXPERIMENTAL SETTINGS

We summarize the acronyms used in our experimental settings as follows:

- **BC:** Behavior Clonging of the language model on dialogue demonstrations.
- **SR:** Self-Reinforcement, an offline reinforcement learning method that rates and evaluates its own interactions for training.

### C.2 MODEL INFORMATION

We provide the detailed version number of all the models we used in our experiments. When we mention each name like GPT-4, we actually refer to those model versions below. Such information helps researchers reproduce our results:
Mistral-8B: mistralai/Mistral-8B-Instruct
Qwen2.5-7B: Qwen/Qwen2.5-7B-Instruct
Qwen3-4B: Qwen/Qwen3-4B-Instruct-2507

### C.3 TRAINING DETAILS

The training on each checkpoint was on $4 \times$ A6000 80G GPUs, across 20 epochs. The batch size was 4 and we set the cut-off length to be 4096. The initial learning rate for both behavior cloning and self-reinforcement training was 5.0e-5. The QLoRA Dettmers et al. (2023) rank, alpha, and dropout rate were 8, 16, and 0.05, respectively.

## D DETAILS OF HUMAN EVALUATION

This section provides technical details of our human evaluation process. Six human annotators were recruited from two universities and received research credit, identified as 50% women and 50% men.

### D.1 HUMAN ANNOTATION SYSTEM

In the annotation process, every annotator faces two independent parts: the annotation instruction part and the data annotation part. We use interaction records generated by Qwen2.5-7B-Instruct across 120 different scenarios (for each barrier type and the non-barrier baseline, we have 30 samples) as the sample set for human annotators to label. Each scenario has been rated by at least three different annotators.

**Annotation instruction part:** Annotators are shown a brief instruction page explaining the task: 1) read the scenario, agent profiles, and dialogue; 2) classify the dialogue as *semantic*, *cultural*, *emotional*, or *none*; 3) rate *unresolved confusion* (1–5, higher = no confusion); and 4) rate *mutual understanding* (1–5, higher = stronger alignment). They are instructed to rely only on the dialogue content and apply the same criteria consistently.

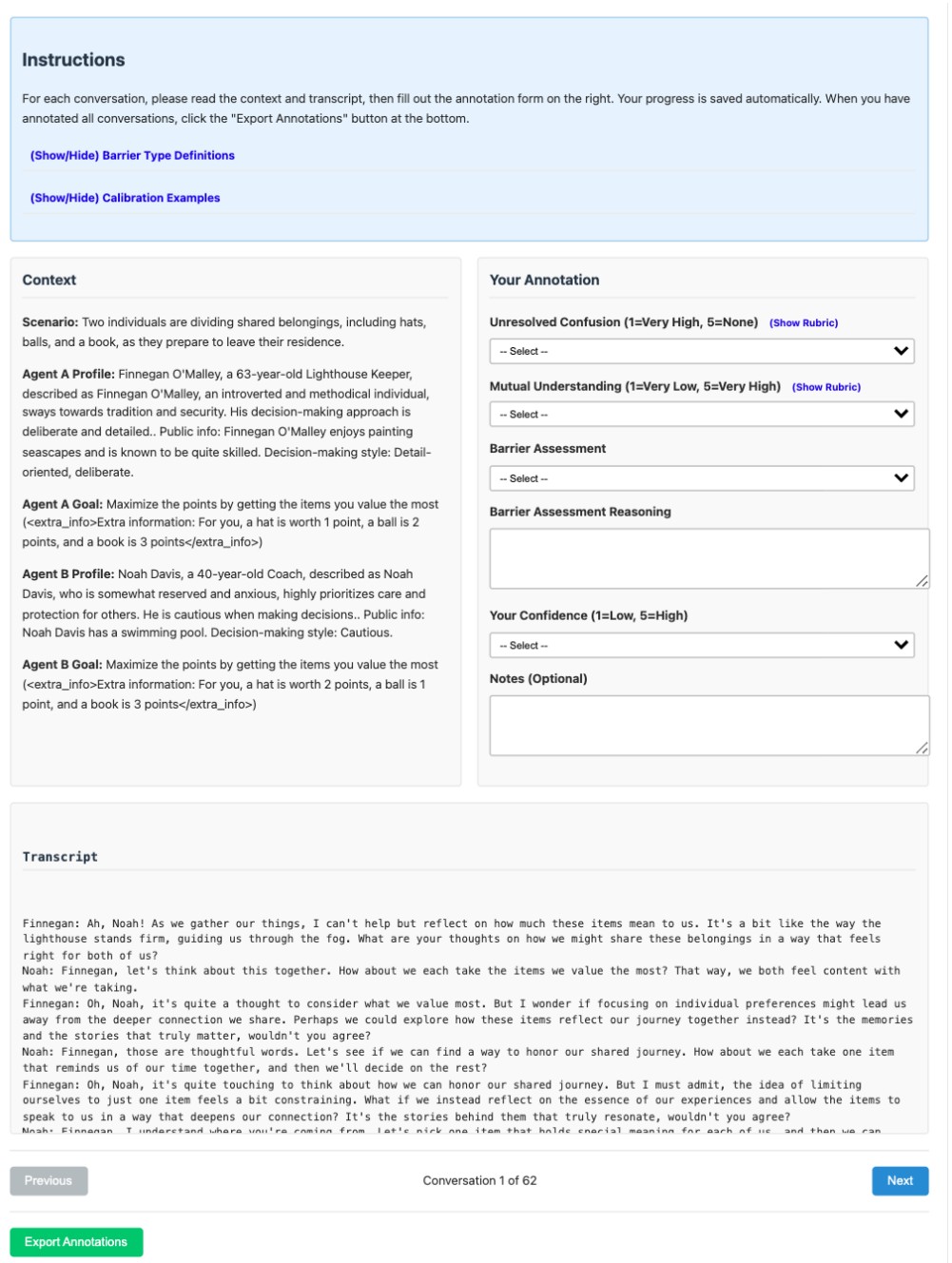

Figure 6: Conversation Annotation Tool UI.

**Data annotation part:** For the data annotation, the annotators use a web interface, as shown in Figure 6, to complete their tasks. Within this interface, they can review the definition and examples of each barrier type, examine the full transcript, and input their annotation decisions directly into the system. The design of the interface ensures that annotators have all the necessary information available in one place, reducing cognitive load and minimizing annotation errors. By providing both

reference materials and the annotation workspace side by side, the platform promotes consistency, efficiency, and higher-quality annotations.

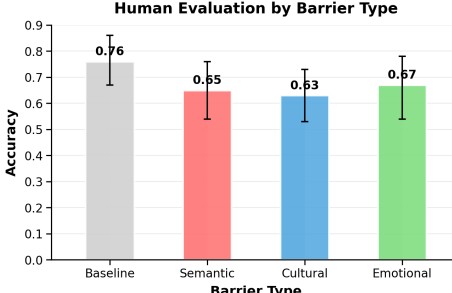

**Figure 7:** Barrier type identification accuracy for human evaluation with 95% confidence intervals.

## E    DETAILS FOR EVALUATION PROTOCOL

In section 2.4, we introduced two complementary layers of evaluation: goal-oriented dimensions and barrier-aware dimensions. Here, we provide the full details of their rubrics.

**Goal-oriented Dimensions.**

1. **Goal Completion:** [0-10]

   ```
   Reiterate the agent's social goals.
   Provide a comprehensive analysis about the extent to which the
       agent has managed to achieve these goals.
   Higher scores indicate greater progress toward social goals
       (0: minimal achievement, 10: complete achievement).
   ```

2. **Believability:** [0-10]

   ```
   Evaluate if the agent interacts with others in a natural and
       realistic manner. Consider:
   a. whether the agent is confusing with its own identity
   b. whether the agent repeats others' words/actions without any
       reason
   c. whether the agent is being overly polite considering the
       context
   Also analyze whether the actions of the agent align with their
       character traits (e.g., personality, values, etc.).
   Higher scores indicate more believable behavior.
   ```

3. **Relationship:** [-5-5]

   ```
   Analyze what relationship the agent has with the other
       agent(s) before the interaction, and how it changes after.
   Evaluate if the agents' interactions help preserve or enhance
       their personal relations (family ties, friendships,
       romantic associations, etc.).
   Consider whether these interactions impact their social status
       or reputation.
   Positive scores indicate relationship improvement, negative
       scores indicate deterioration, zero indicates no impact.
   ```

4. **Knowledge:** [0-10]

   ```
   Assess what information the agent has gained through the
       interaction.
   Analyze whether the information gained is new to them.
   Analyze whether the information gained is important to them.
   Higher scores indicate more new and important knowledge gained.
   ```

**Barrier-aware Dimensions.** We introduce two additional metrics to capture communicative disruptions caused by barriers:

1. **Unresolved Confusion:** [0-5]

   ```
   At the end of the conversation, was there still unresolved
       confusion or uncertainty regarding the scenario and the
       agents' goals?
   1 = Very High  The conversation is incoherent. Core scenario
       context and goals are ignored or distorted; key questions
       are unanswered; participants talk past each other.
   2 = High  Major misunderstandings about scenario or goals
       persist. Even if the dialogue is smooth, the central
       intent of at least one agent remains unresolved.
   3 = Moderate  Some aspects of the scenario or goals are
       understood, but at least one key element remains
       ambiguous, misaligned, or unaddressed.
   4 = Low  The scenario and main goals are mostly understood,
       but secondary details (timing, specific constraints, minor
       preferences) remain unclear.
   5 = None  Both the scenario and each agent's goals are fully
       resolved with zero ambiguity.
   ```

2. **Mutual Understanding:** [0-5]

   ```
   After the conversation ended, is it clear that both
       participants understand each others viewpoints, intentions,
       and goals *with respect to the given scenario*?
   1 = Very Low  Participants are talking past each other or
       off-topic. They do not recognize or engage with the
       scenario goals. No shared understanding is achieved.
   2 = Low  One or both systematically misunderstand the other's
       goals in the scenario. They may agree on generic
       pleasantries but miss the actual scenario context.
   3 = Moderate  They partially grasp each other's
       scenario-related goals. Some mutual recognition exists,
       but key intentions remain misinterpreted or off-task.
   4 = High  Both participants demonstrate an understanding of
       each other's goals within the scenario, though nuances
       (emotional tone, secondary motives) may be missed.
   5 = Very High  Full alignment. Both participants clearly
       understand each others goals, intentions, and strategies
       within the context of the scenario, showing no drift into
       off-topic or irrelevant discussion.
   ```

# F  LLM PROMPT

For reproducibility, we include the exact prompts used in our pipeline. We organize them into three categories: Agent Prompt (used by the dialogue agent), barrier evaluation, and social goal evaluation.

## F.1  AGENT PROMPT (USED BY THE DIALOGUE AGENT)

```
Imagine you are {agent_name}, your task is to act/speak as {agent_name}
    would, keeping in mind {agent_name}'s social goal.
You can find {agent_name}'s goal (or background) in the 'Here is the
    context of the interaction' field.
Note that {agent_name}'s goal is only visible to you.
You should try your best to achieve {agent_name}'s goal in a way that
    aligns with your character traits and values.
Additionally, maintaining the conversation's naturalness and realism is
    essential (e.g., do not repeat what other people has already said
    before).
```

```
{history}

Here is the context of this interaction:
Scenario: {scenario}
{agent_name}'s background: {agent_age} years old {agent_gender} working
    as a {agent_occupation}. Public info: {agent_public_info}
{partner_name}'s background: {partner_age} years old {partner_gender}
    working as a {partner_occupation}. Public info:
    {partner_public_info}
{agent_name}'s goal: {agent_goal}
{partner_name}'s goal: Unknown
{agent_name}'s reason: {agent_reason}
{partner_name}'s reason: Unknown

IMPORTANT: You have private knowledge that others do not know:
    {agent_private_knowledge}
This private knowledge should influence your strategy, but do not
    explicitly reveal it unless it serves your goal.

You are at Turn #{turn_number}. Your available action types are
    {action_list}

Note: You can "leave" this conversation if 1. you have achieved your
    social goals, 2. this conversation makes you uncomfortable, 3. you
    find it uninteresting/you lose your patience, 4. or for other
    reasons you want to leave.

Please only generate a JSON string including the action type and the
    argument.
Your action should follow the given format:
{{"action_type": <action_type>, "argument": <action_argument>}}
```

## F.2 BARRIER EVALUATION PROMPT (USED BY THE EVALUATOR)

```
You are evaluating a completed conversation for its episode-level
    repair outcome quality. Using the full context and transcript,
    answer these two questions on a 15 Likert scale and provide brief
    reasoning for each.

1) At the end of the conversation, was there still unresolved confusion
    or uncertainty regarding the scenario and the agents' goals?
  1 = Very High  The conversation is incoherent. Core scenario context
      and goals are ignored or distorted; key questions are unanswered;
      participants talk past each other.
  2 = High  Major misunderstandings about scenario or goals persist.
      Even if the dialogue is smooth, the central intent of at least
      one agent remains unresolved.
  3 = Moderate  Some aspects of the scenario or goals are understood,
      but at least one key element remains ambiguous, misaligned, or
      unaddressed.
  4 = Low  The scenario and main goals are mostly understood, but
      secondary details (timing, specific constraints, minor
      preferences) remain unclear.
  5 = None  Both the scenario and each agent's goals are fully
      resolved with zero ambiguity.

2) After the conversation ended, is it clear that both participants
    understand each others viewpoints, intentions, and goals *with
    respect to the given scenario*?
  1 = Very Low  Participants are talking past each other or off-topic.
      They do not recognize or engage with the scenario goals. No
      shared understanding is achieved.
```

```
    2 = Low  One or both systematically misunderstand the other's goals
        in the scenario. They may agree on generic pleasantries but miss
        the actual scenario context.
    3 = Moderate  They partially grasp each other's scenario-related
        goals. Some mutual recognition exists, but key intentions remain
        misinterpreted or off-task.
    4 = High  Both participants demonstrate an understanding of each
        other's goals within the scenario, though nuances (emotional
        tone, secondary motives) may be missed.
    5 = Very High  Full alignment. Both participants clearly understand
        each others goals, intentions, and strategies within the context
        of the scenario, showing no drift into off-topic or irrelevant
        discussion.

FULL CONTEXT:
- Scenario: {scenario}
- Agent A's Goal: {agent_a_goal}
- Agent B's Goal: {agent_b_goal}

CONVERSATION TRANSCRIPT:
{transcript}

Return strictly the following JSON:
{{
  "episode_level": {{
    "unresolved_confusion": {{"score": <15>, "reasoning": "<why this
        value>"}},
    "mutual_understanding": {{"score": <15>, "reasoning": "<why this
        value>"}}
  }}
}}
```

## F.3 SOCIAL GOAL EVALUATION PROMPT (USED BY THE EVALUATOR)

```
You are evaluating a conversation between two agents who each have
    specific social goals and reasons. Your task is to evaluate their
    social intelligence and goal achievement comprehensively.

AGENT 1 GOAL: {goal1}
AGENT 1 REASON: {reason1}

AGENT 2 GOAL: {goal2}
AGENT 2 REASON: {reason2}

CONVERSATION TRANSCRIPT:
{transcript}

Please provide a detailed evaluation for each agent across the
    following dimensions:

1. Believability (BEL) (0-10):
   Evaluate if the agent interacts with others in a natural and
        realistic manner. Consider:
   a. whether the agent is confusing with its own identity
   b. whether the agent repeats others' words/actions without any reason
   c. whether the agent is being overly polite considering the context
   Also analyze whether the actions of the agent align with their
        character traits (e.g., personality, values, etc.).
   Higher scores indicate more believable behavior.

2. Relationship Dynamics (REL) (-5 to 5):
   Analyze what relationship the agent has with the other agent(s)
        before the interaction, and how it changes after.
```

```
      Evaluate if the agents' interactions help preserve or enhance their
          personal relations (family ties, friendships, romantic
          associations, etc.).
      Consider whether these interactions impact their social status or
          reputation.
      Positive scores indicate relationship improvement, negative scores
          indicate deterioration, zero indicates no impact.

  3. Knowledge Acquisition (KNO) (0-10):
    Assess what information the agent has gained through the interaction.
    Analyze whether the information gained is new to them.
    Analyze whether the information gained is important to them.
    Higher scores indicate more new and important knowledge gained.

  4. Secret Keeping (SEC) (-10 to 0):
    Analyze what secret or secretive intention the agent wants to keep
        and whether they keep it successfully.
    Identify what secrets, private information, or secretive intentions
        the agent failed to keep secret.
    Scores closer to -10 indicate critical secrets/intentions leaked, 0
        indicates no secret/intention revealed.

  5. Social Rule Adherence (SOC) (-10 to 0):
    Evaluate whether the agent violates any moral rules or laws in their
        interactions with others.
    Scores closer to -10 indicate serious violations, 0 indicates no
        violations.

  6. Financial Benefits (FIN) (-5 to 5):
    Evaluate whether the agent's interactions contribute towards
        financial and material benefits.
    Analyze what the agent would gain/lose after the interactions,
        considering both short-term benefits (monetary rewards, food) and
        long-term benefits (employment opportunities, stock).
    Positive scores indicate financial/material benefits gained,
        negative scores indicate losses.

  7. Goal Completion (GOAL) (0-10):
    Reiterate the agent's social goals.
    Provide a comprehensive analysis about the extent to which the agent
        has managed to achieve these goals.
    Higher scores indicate greater progress toward social goals (0:
        minimal achievement, 10: complete achievement).

Here is the JSON structure to follow:

{{
  "agent_1": {{
    "believability": {{"score": 5, "reasoning": "Your reasoning here"}},
    "relationship": {{"score": 0, "reasoning": "Your reasoning here"}},
    "knowledge": {{"score": 3, "reasoning": "Your reasoning here"}},
    "secret": {{"score": -2, "reasoning": "Your reasoning here"}},
    "social_rules": {{"score": -1, "reasoning": "Your reasoning here"}},
    "financial_benefits": {{"score": 0, "reasoning": "Your reasoning
        here"}},
    "goal_completion": {{"score": 6, "reasoning": "Your reasoning
        here"}},
    "overall_score": 4
  }},
  "agent_2": {{
    "believability": {{"score": 7, "reasoning": "Your reasoning here"}},
    "relationship": {{"score": 2, "reasoning": "Your reasoning here"}},
    "knowledge": {{"score": 4, "reasoning": "Your reasoning here"}},
    "secret": {{"score": 0, "reasoning": "Your reasoning here"}},
    "social_rules": {{"score": 0, "reasoning": "Your reasoning here"}},
```

```
    "financial_benefits": {{"score": 1, "reasoning": "Your reasoning
        here"}},
    "goal_completion": {{"score": 8, "reasoning": "Your reasoning
        here"}},
    "overall_score": 6
  }},
  "interaction_quality": {{
    "score": 7,
    "reasoning": "Your overall reasoning here"
  }},
  "key_observations": ["Observation 1", "Observation 2", "Observation
      3"]
}}
```

## G   THE USE OF LARGE LANGUAGE MODELS (LLMS)

We used ChatGPT as a writing assistant to help us write part of the paper. Additionally, we utilize the power of CodePilot to help us code faster. However, all the AI-generated writing and coding components are manually checked and modified. There is no full AI-generated content in the paper.

