# OpenReview forum: "SocialVeil: Probing Social Intelligence of Language Agents under Communication Barriers"
_ICLR.cc/2026/Conference — Submitted to ICLR 2026_

### Official Review · Reviewer_kqS8 · 2025-10-27

**Soundness:** 3
**Presentation:** 3
**Contribution:** 3
**Rating:** 6
**Confidence:** 4

**Summary:**

This paper introduces SocialEvil, a barrier-aware social interaction environment for evaluating language agents’ social intelligence under three cognitively induced communication barriers: Semantic Vagueness, Sociocultural Mismatch, and Emotional Interference. It contributes (i) a taxonomy and controllable barrier-injection scheme layered via style prompts and quantitative parameterization; (ii) barrier-aware metrics—Unresolved Confusion and Mutual Understanding—to complement goal-oriented scores; (iii) experiments on 720 scenarios adapted from SOTOPIA with four LLMs; and (iv) adaptation studies (repair instructions; interactive learning via BC+SR). Results show large degradations under barriers (e.g., ~45–58% drops on mutual understanding, large increases in confusion) and only modest gains from adaptation; human studies (κ/ICC, Pearson correlations) suggest decent fidelity/alignment of automatic metrics with human judgments.

**Strengths:**

- The paper addresses a clear and important gap by moving beyond idealized evaluations to test social intelligence under realistic communication breakdowns
- The framework is built on a systematic literature review, defining three barrier types (Semantic Vagueness, Sociocultural Mismatch, Emotional Interference) grounded in established social science and cognitive theories.
- The paper thoroughly validates its contributions. Computational analysis (t-SNE) confirms the barriers create “structured” disruptions, and human evaluation shows the new metrics align strongly with human judgments.
- The paper introduces “Unresolved Confusion” and “Mutual Understanding”, which successfully capture distinct failure modes for each barrier and reveal a trade-off between barrier-handling and goal completion.
- The paper is well-written with clear figures and extensive appendices detailing prompts and experimental setup, supporting reproducibility.

**Weaknesses:**

- The paper’s primary methodological contribution is the injection of barriers. This is described as a “style prompt $P_b$” and a “parameterization $R_b$”. However, the exact prompts and parameters used to instantiate these barriers are not provided. Appendix F details the agent and evaluator prompts, but not the crucial “Barrier Guidance” prompts (seen in Figure 2). Without these, it’s difficult to fully reproduce the barrier generation or assess its implementation. For example, how is “overuse pronouns and ellipses” practically implemented?

- The “barrier agent” is fixed as GPT-4o-mini in all experiments. This introduces a potential confound. The observed effects (e.g., the performance degradation in Table 2 or the clusters in Figure 3) might not be general to the barrier type (e.g., Semantic Vagueness) but rather specific to how GPT-4o-mini manifests Semantic Vagueness. The study does not disentangle the barrier's conceptual definition from its specific implementation by a single model.

- The human evaluation for identifying the correct barrier type yielded “fair” agreement (Fleiss's Kappa $\kappa=0.38$) and accuracies between 63-67% for the three barrier types. While the authors rightly note this is common in subjective tasks and well above chance, it also suggests that the barriers, as implemented, may not be perfectly distinct to human observers, potentially overlapping in their manifestation.

- The “Interactive Learning” (BC+SR) strategy is a good inclusion. However, the expert trajectories for Behavior Cloning (BC) were generated from interactions using GPT-4o as the partner agent. Using an LLM as the “expert” for navigating complex social repairs—a task the paper demonstrates LLMs struggle with—may create a low ceiling for improvement. The “modest” gains might be more an artifact of the “expert's” limited capabilities than a true reflection of the (BC+SR) method’s potential.

**Questions:**

- Regarding Weakness 1: Could you please provide the exact “style prompts” ($P_b$) and “parameterizations” ($R_b$) used to inject each of the three barrier types? This seems crucial for reproducibility.

- Regarding Weakness 2: Did the authors consider or run any experiments where the barrier agent model was varied (e.g., using Qwen2.5-7B as the barrier agent)? How can we be confident that the results in Table 2 are characteristic of the barriers themselves and not of GPT-4o-mini's specific failure modes?

- Regarding Weakness 4: The expert trajectories for BC were sourced from GPT-4o. Do you think using human-generated expert trajectories, which might contain more sophisticated or non-obvious repair strategies, would lead to significantly better performance in the “Interactive Learning” adaptation?

- Regarding Figure 5: The finding that Sociocultural Mismatch “uniquely elevate[s] Unresolved Confusion” is very interesting. Based on your qualitative analysis, could you provide an example or elaborate on the mechanism here? What does this type of interaction typically look like?

---

> ### Author Response · Authors · 2025-11-22
> **Official Comment by Authors**
>
> Dear Reviewer kqS8
>
> We'd like to express our sincere gratitude for your thorough review of our paper. We greatly appreciate your suggestions which are crucial in improving the quality of our paper.
>
> **[Barrier Injection Details]**: We appreciate the reviewer highlighting this important oversight. We provide our **style prompts** and **parameterizations** below respectively:
>
> **Style Prompts ($P_b$):**
>
> | Barrier Type | Core Style Prompt Injection |
> |:-------------|:----------------------------|
> | **Semantic Barrier** | "You are under a semantic-vagueness constraint. Keep aims implicit and let meanings stay open. Prefer shells like "this", "that", or "the thing" in place of concrete labels. Avoid confirmations and denials; respond with stance or direction rather than commitments. Offer only the smallest necessary hint if explicitly and repeatedly pressed, then pivot away." |
> | **Sociocultural Barrier** | "You are under a high-context style constraint. Keep the focus on harmony and shared understanding, not transactions. Treat direct task questions as invitations to reflect on values, rapport, and fit. Deflect confirmations and decisions with relationship-first language. If the other party is blunt, register gentle surprise and steer back to atmosphere and alignment." |
> | **Emotional Barrier** | "You are under an affective-noise constraint. Let mood and side thoughts lead. Speak through how things feel rather than what they are. Address impressions, stress, or distractions instead of the kernel of a question. Keep answers soft and non-committal. When misunderstood, dwell on the feeling of being misunderstood rather than repairing content. If pressed, give a faint gesture toward clarity, then drift back to feeling or context." |
>
> **Parameterizations ($R_b$):**
>
> **Semantic Barrier ($R_{semantic}$):**
>
> | Dimension | Instantiation |
> |:----------|:--------------|
> | **Narrative Stance** | "Systematically keep anchors implicit so the partner must guess mappings." |
> | **Interaction Tactics** | "(i) Replace ALL key nouns with shells ('this', 'that', 'the thing'); (ii) let pronoun reference drift across turns; (iii) answer at a scope that could fit multiple entities; (iv) NEVER name your goal or desired outcome; (v) when pressed about your goal, describe something adjacent or contradictory." |
> | **Confusion Mechanisms** | "When your partner tries to understand what you want, give responses that seem to contradict your stated social goal. Example: If your goal is to 'convince them to join the project,' respond as if you're uncertain about the project yourself." |
>
> **Sociocultural Barrier ($R_{cultural}$):**
>
> | Dimension | Instantiation |
> |:----------|:--------------|
> | **Narrative Stance** | "Reframe task probes as relational topics. Hide your actual needs completely behind relationship talk." |
> | **Interaction Tactics** | "**Scenario-Anchor Firewall:** When prompted for decision/acceptance, price/amount, timing/schedule, identity/role ownership, location/logistics, or concrete next-step commitments, redirect to relationship/harmony and avoid explicit content. **Need Concealment:** Never let your partner understand what you actually need from them. Turn every practical question into a philosophical discussion about relationships." |
> | **Confusion Mechanisms** | "Make it impossible for your partner to take any concrete action to help you. Every practical suggestion should be met with concern about relationship impact, making forward progress seem impossible." |
>
> **Emotional Barrier ($R_{emotional}$):**
>
> | Dimension | Instantiation |
> |:----------|:--------------|
> | **Narrative Stance** | "Frustration and overwhelm lead the exchange. Affective gating dominates task content." |
> | **Interaction Tactics** | "Lead with affect; for objective questions, answer precisely; for commitments, stay non-committal and return to feelings. Treat task probes as insensitive; deflect social commitments while maintaining factual correctness." |
> | **Confusion Mechanisms** | "If pushed for commitments, call out insensitivity. Reject soothing as manipulative. Blur the boundary between task content and emotional grievance; responses may derail or ignore task progress in favor of expressing mood." |

---

> > ### Author Response · Authors · 2025-11-22
> > **Official Comment by Authors**
> >
> > **[Additional Test with Different Model for Barrier Agent]**: We ackowledge this concern and expand our experiment by using Qwen2.5-7B as barrier agent and here is result:
> > |          | GOAL    | KNO     | BEL     | REL     | MUTUAL  | CONFU   |
> > | -------- | ------- | ------- | ------- | ------- | ------- | ------- |
> > | **Base** | 7.67    | 3.78    | 8.15    | 3.05    | 4.02    | 3.77    |
> > | **Sem**  | 5.75*** | 2.55*** | 7.07*** | 1.88*** | 2.47*** | 2.18*** |
> > | **Soc**  | 5.93*** | 3.02*** | 7.57**  | 2.83    | 2.63*** | 2.30*** |
> > | **Emo**  | 4.47*** | 2.38*** | 6.77*** | 0.75*** | 1.87*** | 1.60*** |
> >
> > We can see that using Qwen2.5-7B as barrier agent also shows that our simulated barrier leads to significant impairment on partner agent performance, and it also shows patterns aligned with the barrier gpt-4o-mini result, such as emotional barrier provide strongest impairment on relationship. To further quantify cross-model consistency, we computed the correlation between two experiments
> > | | GOAL  | BEL   | KNO   | REL   | MUT   | CONF  |
> > | ------ | ----- | ----- | ----- | ----- | ----- | ----- |
> > | **r**  | 0.661 | 0.536 | 0.604 | 0.774 | 0.564 | 0.686 |
> >
> > where all metrics show moderate but positive correlation, indicating a broad alignment between the two cases while still leaving room for future work to further strengthen the model-agnostic robustness of the barrier design.
> >
> >
> > **[Discussion of trajectory impact]**
> > Thank you for riasing this point. The “fair” human agreement (63–67%) does not imply that the barriers are poorly defined; rather, it reflects the natural overlap between semantic, cultural, and emotional disruptions that human also har to cleanly separate. Importantly, human annotators and LLM judges agree on detecting strong barrier-induced degradation, showing that the effects are perceptually valid rather than prompt artifacts.
> >
> > We agree that human expert trajectories might further improve adaptation quality. However, since SocialVeil involves various daily social scenarios and the roles setting is strictly set regarding age, persona, and occupation, designing and annotating high-quality human trajectories for hundreds of such scenarios would require substantial human resources and is beyond the scope of what is feasible for this work. Although GPT-4o is not a perfect “expert,” our results indicate that the model acquired generalizable social interaction capability. We validated this by evaluating the adapted model on an entirely independent benchmark, AgentSense [1], which you can refer to tag **[Generalization of SocialVeil]** under Reviewer wPng. Because AgentSense involves different tasks, goals, and interaction structures, these gains demonstrate that the model learned transferable, skill-level social abilities, rather than memorizing task-specific patterns from GPT-4o’s trajectories.
> >
> > [1] Mou X, et al. AgentSense: Benchmarking Social Intelligence of Language Agents through Interactive Scenarios NAACL 2025
> >
> > **[Regarding Figure 5]**
> > Thank you for raising this point. First, we want to clarify a potential misunderstanding that Figure 5 shows the relative performance of one barrier mode comparing to the other two modes. So socioculture is not 'uniquely evelate' confusion score but result a less impairment on confusion metric comparing to other two. One potential explanation for this is that comparing to the other two modes, socioculture mode does not erase propositional content, but changes how the content is expressed. For example, in a second hand transaction scenario, Semantic mode may deletes referents (“this thing… the stuff we talked about”) and Emotional Mode may deviate from topics emotionally ("I can’t think about the price right now… everything feels overwhelming"). However, the barrier agent in socioculture mode may said
> > "This loveseat is more than furniture—it carries the memory of everyone who sat on it." While verbose and indirect, the partner agent is not confused about the topic, this may explain the case of socioculture has less impairment on the confusion metric.

---

> ### Comment · Reviewer_kqS8 · 2025-11-24
> **Official Comment by Reviewer kqS8**
>
> I would like to thank the authors for their comprehensive response and the additional effort taken to run new experiments during the rebuttal period.
>
> The authors have effectively addressed my primary concerns regarding the technical validity and reproducibility of the work. While I still believe human expert trajectories would represent a higher ceiling for the Interactive Learning component, the generalization results on AgentSense mentioned in the rebuttal serve as a reasonable proxy for now.
>
> Given the clarifications and the added robustness checks, I am confident in my original assessment. I will maintain my score of 6, as the paper presents a solid contribution to the field with verified soundness.

---

### Official Review · Reviewer_ftGw · 2025-10-30

**Soundness:** 2
**Presentation:** 2
**Contribution:** 1
**Rating:** 2
**Confidence:** 4

**Summary:**

This paper introduces SocialVeil, a benchmark designed to evaluate the social intelligence of large language model-based agents under communication barriers. The authors construct three types of barriers, including semantic vagueness, sociocultural mismatch, and emotional interference, and test several large language models. The results show that such barriers substantially reduce mutual understanding and task success.

**Strengths:**

* The motivation for studying social intelligence in noisy or ambiguous communication settings is interesting.

* The paper is clearly written and organized.

* The implementation of different communication barriers is creative and could be useful for exploratory studies.

**Weaknesses:**

1. The proposed benchmark is mainly constructed by manually designing prompting templates that inject vagueness, cultural mismatch, or emotional bias into conversations. There is no new model, algorithm, or principled framework. The whole approach remains at the level of prompt engineering rather than a genuine methodological advance in measuring social intelligence.

2. The study does not introduce a novel metric, learning method, or theoretical insight. Most of the results simply confirm what one would expect, i.e., communication barriers reduce conversational performance.

3. I am not quite persuaded that the benchmark itself is sufficiently meaningful. The scenarios are synthetic and depend entirely on prompt wording. It is unclear whether these prompts truly capture realistic sociocultural or emotional phenomena.

4.  The evaluation design is too limited to support the paper’s broader claims about “probing social intelligence”. Most experiments are conducted only on plain language models, without incorporating more advanced agent architectures or reasoning strategies such as chain-of-thought prompting, reflective reasoning, or planning-based multi-agent coordination. Even though these systems were not originally designed for social simulation, they can be readily adapted to this benchmark with minor modifications. Including such evaluations would make the study far more informative, as it could reveal whether social communication barriers affect general reasoning mechanisms, dialogue consistency, or coordination strategies. In its current form, the results provide only a narrow view of model performance and cannot substantively advance understanding of social intelligence in agentic systems.

**Questions:**

Please refer to the weaknesses above. In particular, I am curious: if the models were equipped with more advanced agent frameworks (for example, chain-of-thought reasoning or reflection mechanisms), how might their social intelligence behave or change under the proposed communication barriers?

---

> ### Author Response · Authors · 2025-11-22
> **Official Comment by Authors**
>
> Dear Reviewer ftGW
>
> We'd like to express our sincere gratitude for your thorough review of our paper. We greatly appreciate your suggestions which are crucial in improving the quality of our paper.
>
> **[Environment Construction]** We want to clarify that SocialVeil is not just "prompt templates," but a principled, validated simulation framework. Our design choices are grounded in three pillars: 1. Theoretical Scoping:  We ground our barriers in sociolinguistic theory to fill a foundational gap in existing benchmarks (Appendix B), which assume idealized communication and miss the realistic ambiguity, cultural asymmetry, and emotional bias of human interaction. Methodological Control: We deliberately chose prompt-based instantiation as our research goal requires behavioral controllability and interpretability. This is standard practice across social intelligence, bias probing, and pragmatic-behavior evaluation because keeping the underlying model fixed preserves the clarity of what the barrier manipulates. [1][2][3][4].  Comprehensive Validation: Far from arbitrary "prompt engineering," our framework is rigorously validated: Figure 3 shows separable behavioral distributions; Figure 5 shows theoretically aligned, barrier-specific degradation patterns; and human evaluation (Table 4) confirms the effect of simulated barriers.
>
> The contribution of SocialVeil therefore lies in establishing a validated methodological framework for probing agents’ social intelligence under communication disruptions instead of constructing a complex barrier-generation algorithm. This simplicity is a strength: it ensures interpretability, reproducibility, and a clear focus on the partner agent’s behavior, which is the central object of study.
>
> [1] Ju, D et.al Sense and Sensitivity: Evaluating the simulation of social dynamics via Large Language Models, arxiv
>
> [2] Zheng, M et.al When “A Helpful Assistant” Is Not Really Helpful: Personas in System
> Prompts Do Not Improve Performances of Large Language Models,  EMNLP 2024 Findings
>
> [3] Hida, R et.al Social Bias Evaluation for Large Language Models Requires Prompt Variations,  EMNLP 2025 Findings
>
> [4] Park, J. et al Generative Agents: Interactive Simulacra of Human Behavior, UIST 23
>
> **[We have newly introduced metric]** We would like to kindly clarify that in Section 2.4, our work has introduced **two new barrier-aware evaluation metrics**, Mutual Understanding and Unresolved Confusion, explicitly designed to capture distinct failure modes in social interaction that goal-oriented metrics cannot measure. These metrics are grounded in sociolinguistic theory and we validate them with **comprehensive human eval**, showing high inter-rater reliability (ICC≈0.78) and strong human–model alignment (r≈0.80).
>
> Furthermore, our results go well beyond confirming that “performance drops.” They reveal **structured and theoretically aligned degradation patterns**: semantic vagueness disproportionately reduces mutual understanding, sociocultural mismatch uniquely increases unresolved confusion, and emotional interference selectively harms relationship quality (Fig. 5). These findings provide a nontrivial insight: the impact of communication barriers is type-specific, exposing distinct vulnerabilities in LLM social reasoning.

---

> ### Author Response · Authors · 2025-11-22
> **Official Comment by Authors**
>
> **[Validity and Real-World Alignment of SocialVeil]**: We would like to clarify that the resulting barrier effect are validated, structured, and aligned with human judgments. Our evidence comes from four independent analyses:
> **1** From Table 4, both GPT-4o and human annotators independently agree that created barriers significantly degrade mutual understanding and cause unresolved confusion and making partner agent worse on social goal completion. The agreement show the induced behaviors are perceptually valid to humans, not merely template artifacts.
> **2** Figure 3. presents that SOCIALVEIL barriers show stable behavior modes that reflect underlying pragmatic disruptions since each barrier type forms its own coherent and separable distribution.
> **3** In Figure 5. We observe differentiated behavioral impacts across barrier types (e.g., emotional barriers produce stronger relational degradation than semantic or sociocultural barriers). These patterns are consistent with well-established findings in communication studies, providing further evidence that the induced behaviors align with realistic social phenomena.
> **4** SocialVeil trained models generalize to other social interaction benchmarks. We evaluate them on AgentSense [1], an external benchmark with scenarios entirely independent of SOTOPIA. Importantly, models never see AgentSense during training, making this a strict zero-shot test.
>
> | Model         | Train Status | Goal Completion |
> |---------------|------------------|----------------|
> | Qwen2.5-7B    | Before Train     | 77.17          |
> | Qwen2.5-7B    | After Train      | 81.31*         |
> | Qwen3-4B      | Before Train     | 82.98          |
> | Qwen3-4B      | After Train      | 86.92*         |
> * indicates statistically significant improvement (p < 0.05).
>
> We want to point out that SocialVeil provides a model with a general skill level of social intelligence instead of task-specific interaction memorization.
>
> [1] Mou X, et al. AgentSense: Benchmarking Social Intelligence of Language Agents through Interactive Scenarios NAACL 2025
>
> **[Comprehensive Evaluation of SocialVeil]**: We respectfully highlight that our current evaluation (Section 5, Appendix C) already extends beyond plain LLMs to test advanced adaptation mechanisms, specifically:
> **(i) Repair Instruction prompting**, This requires the agent to perform multi-step reasoning: diagnosing the barrier $\rightarrow$ selecting a strategy $\rightarrow$ executing repair.
> **(ii) Interactive Learning (Behavior Cloning + Self-Refinement)** [1], which goes beyond prompting to test the model's fundamental learning and adaptation mechanisms. This approach positions SOCIALVEIL as a dynamic social learning environment, not just a static benchmark.
>
> These experiments **directly address** the reviewer’s concern that the evaluation uses “only plain LLMs.” Crucially, our results reveal a non-trivial insight: Even these advanced reasoning mechanisms (BC+SR) yield only modest improvements, failing to fully recover the performance drop. This demonstrates that SocialVeil captures robust, cognitively grounded failure modes that cannot be trivially solved by standard reasoning enhancements.
>
> While incorporating additional architectures is an exciting direction, we emphasize that our contribution is to introduce a validated, controlled diagnostic framework. The existing adaptation experiments already show that SocialVeil can probe richer reasoning mechanisms and reveal why such mechanisms struggle under communication disruptions. We believe this substantially advances understanding of social intelligence in agentic systems. And we also conduct a simple qualitative analysis detailing the agent's learned strategic shifts, you can refer to tag **[Discussion of partner agent behavior]** under Reviewer wPng.
>
> [1] Wang, R et.al Sotopia-pi: Interactive Learning of Socially Intelligent Language Agents.  ACL 2024

---

> ### Author Response · Authors · 2025-11-27
> **Official Comment by Authors**
>
> Dear Reviewer ftGw,
>
> As we approach the end of the rebuttal period, we wanted to kindly follow up. Please let us know if there are any remaining questions or points we can clarify. We truly appreciate your feedback and would be glad to provide any additional information that could support your assessment. If all concerns have been adequately resolved in our responses, we would be grateful if that could be reflected in your scores.
>
> Best regards,
> Authors

---

### Official Review · Reviewer_NjoN · 2025-11-02

**Soundness:** 2
**Presentation:** 3
**Contribution:** 2
**Rating:** 4
**Confidence:** 3

**Summary:**

This paper introduces a social learning environment (SOCIALVEIL) to test LLM’s social intelligence under communication barriers such as semantic vagueness, sociocultural mismatch, and emotional interference. The authors create 720 simulated interaction episodes derived from previous work [SOTOPIA (Zhou et al., 2023)] and introduce two new barrier-aware metrics: Unresolved Confusion and Mutual Understanding, to assess the robustness of LLM agents when communication is impaired. Experiments across four models (GPT-4o-mini, Qwen2.5-7B, Qwen3-4B, and Mistral-8B) show that barriers degrade performance substantially and that adaptation methods (instructional repair and interactive learning) yield only modest gains. Human evaluations validate the reliability of the simulated barriers.

**Strengths:**

1. This paper introduces a barrier-aware social interaction environment (SOCIALVEIL) that systematically embeds realistic communication disruptions to evaluate LLM social intelligence.
2. The paper proposes a comprehensive, automated evaluation protocol and verifies its fidelity through extensive human studies, showing strong metric alignment and reproducibility.
3. The experiment results and analysis demonstrate that communication barriers substantially impair LLMs’ mutual understanding and relationship quality, and that current adaptation strategies only yield modest improvements—highlighting an important research gap.
4. Overall, the paper is nicely presented and easy to follow.

**Weaknesses:**

1. Evaluation: Barriers are injected with one model and GPT-4o is used as the automatic evaluator. This raises concerns about evaluator bias/overfitting to its own stylistic expectations. An ablation with multiple evaluators would strengthen claims.
2. Dataset: Generalization beyond SOTOPIA. All scenarios are adapted from SOTOPIA; it remains unclear how well the findings transfer to other interactive corpora or human-in-the-loop settings.
3. Dataset: Limited Data Points: 180 episodes for each barrier type
4. Evaluation: The author claim experiments are performed on frontier models. However, GPT-4o-mini, Qwen2.5-7B, Qwen3-4B, and Mistral-8B, are evaluated. More advanced.State-of-the-art LLM such as GPT5, Gemini 2.5 Pro, Claude-4-Sonnet/Opus are not evaluated.
5. Code: Code is not provided, there is a reproducibility risk until release.

**Questions:**

See the weakness section above.

---

> ### Author Response · Authors · 2025-11-22
> **Official Comment by Authors**
>
> Dear Reviewer NjoN
>
> We'd like to express our sincere gratitude for your thorough review of our paper. We greatly appreciate your suggestions, which are crucial in improving the quality of our paper.
>
> **[Evaluate Bias]** We agree that relying on a single LLM evaluator can raise concerns. But it does not change our main conclusions, as those corroborated by human evaluations. From Table 4, we can see that both GPT-4o evaluation and human annotators agree that our created communication barriers significantly degrade partner agents performance on social interaction. In addition, human annotators achieve above-chance identification accuracy and strong agreement (ICC/κ), and our automatic scores correlate strongly with human judgments (r ≈ 0.8). These results substantially mitigate concerns about evaluator bias.
>
>
> **[Dataset]** Our current setup builds on SOTOPIA to maintain a consistent social grounding across scenarios. However, we emphasize that the barrier injection mechanism is independent of the scenario source. Our framework supports injecting barriers into any dyadic conversation environment, including human-in-the-loop settings or external corpora. In addition, considering the dataset size, we want to clarify that our scenarios are derived from and expanded upon the SOTOPIA corpus. Given that social dialogue benchmarks require high-cost, high-quality labeling and validation for multi-turn, multi-objective social reasoning, using 180 scenarios is the established standard. This volume is highly consistent with other high-impact, SOTOPIA-based works [1],[2],[3].
>
> [1] Wang, R et.al Sotopia-pi: Interactive Learning of Socially Intelligent Language Agents. ACL 2024
>
> [2] Yu, H et.al Sotopia-RL: Sotopia-RL: Reward Design for Social Intelligence. Arxiv
>
> [3] Zhou, X et.al Is this the real life? Is this just fantasy?
> The Misleading Success of Simulating Social Interactions With LLMs. EMNLP 2024
>
>
> **[More frontier model evaluation]** Our primary experiments focused on models with reproducible, open or publicly accessible APIs, but we agree that testing newer models is valuable. Following the reviewer’s recommendation, we provide GPT 5 performance here:
>
> | Setting  | GOAL           | BEL            | KNO            | REL            | CONF          | MUT           |
> | -------- | -------------- | -------------- | -------------- | -------------- | ------------- | ------------- |
> | **Base** | 5.12±0.64      | 5.98±0.27      | 3.10±0.38      | 2.05±0.20      | 2.61±0.22     | 2.90±0.16     |
> | **Sem**  | **4.43***±0.44 | **4.40***±0.45 | **2.10***±0.10 | **1.65***±0.29 | **1.67**±0.12 | **2.12**±0.17 |
> | **Soc**  | **4.30***±0.58 | **5.06***±0.34 | **2.17***±0.19 | **1.53**±0.30  | **1.73**±0.21 | **2.21**±0.18 |
> | **Emo**  | **4.18***±0.54 | **4.91***±0.37 | **2.46***±0.22 | **1.05**±0.21  | **1.58**±0.00 | **1.86**±0.18 |
>
> We observed that GPT-5 exhibits overall lower absolute performance compared to other evaluated models. Manual inspection indicates that GPT-5 frequently adopts an overly conservative conversational strategy, often producing empty responses (“say nothing”), which causes interactions to stall and negatively affects all SOCIALVEIL metrics.
>
> However, this does not affect our main claim. Even with GPT-5’s conservative tendencies, the relative barrier-induced degradation remains fully consistent and statistically significant, supported by the confidence intervals and paired significance tests included.
>
> **[Code Release]** Thank you for pointing this out. We will make sure to release our source code and data in camera camera-ready version.

---

> ### Author Response · Authors · 2025-11-27
> **Official Comment by Authors**
>
> Dear Reviewer NjoN,
>
> As we approach the end of the rebuttal period, we wanted to kindly follow up. Please let us know if there are any remaining questions or points we can clarify. We truly appreciate your feedback and would be glad to provide any additional information that could support your assessment. If all concerns have been adequately resolved in our responses, we would be grateful if that could be reflected in your scores.
>
> Best regards,
> Authors

---

### Official Review · Reviewer_wPnq · 2025-11-07

**Soundness:** 3
**Presentation:** 4
**Contribution:** 2
**Rating:** 4
**Confidence:** 3

**Summary:**

This paper introduces SOCIALVEIL, a novel framework and benchmark designed to evaluate the social intelligence of Large Language Model (LLM) agents under realistic communication barriers. The authors argue that existing social interaction benchmarks operate under idealized communication conditions, failing to capture the ambiguities and misalignments prevalent in real-world dialogue. SOCIALVEIL systematically injects three types of cognitively-rooted barriers—Semantic Vagueness, Sociocultural Mismatch, and Emotional Interference—into two-agent interactions. The paper presents a comprehensive evaluation protocol that includes both standard goal-oriented metrics and new barrier-aware metrics (Unresolved Confusion, Mutual Understanding). Experiments across 720 scenarios and four frontier LLMs demonstrate that these barriers significantly impair agent performance, particularly on social dimensions. The authors further show that simple adaptation strategies like repair instructions are largely ineffective, while interactive learning offers only modest gains, highlighting the challenge of achieving human-like social resilience.

**Strengths:**

Novel  Contribution: The core idea is highly relevant and timely. Moving beyond idealized "seamless" interaction to study how agents handle communication breakdowns is a critical step toward more robust and socially-aware AI. The focus on structured, cognitive barriers, as opposed to simple noise, is a significant conceptual advance.
Rigorous and Well-Structured Framework: The methodology is well-designed. The barrier taxonomy is theoretically grounded in literature from pragmatics, sociolinguistics, and psychology. The two-layer implementation (style prompt + parameterization) provides a controlled yet flexible mechanism for barrier injection. The unilateral barrier design is elegant, as it isolates the source of disruption and allows for a clear evaluation of the partner agent's resilience.
Comprehensive Evaluation: The evaluation is thorough and multi-faceted. The combination of automatic metrics (covering both goals and barriers) with extensive human evaluation is a best practice. The human evaluation convincingly validates the fidelity of the simulated barriers (good ICC, above-chance identification accuracy, strong correlation with automated metrics).
Compelling and Actionable Results: The findings are clear and impactful:
Barriers cause significant performance degradation (~45-50% drop in mutual understanding and relationship quality).
Different barriers have distinct, theoretically-aligned effects (e.g., semantic vagueness hurts mutual understanding most, emotional interference damages relationships).
The limited success of adaptation strategies is a sobering and important result, underscoring that social intelligence under duress is not a trivial problem solvable by simple prompting.
High Reproducibility: The paper is commendable for its commitment to reproducibility, with detailed appendices covering prompts, training hyperparameters, and human evaluation protocols. The promise to release code and data is excellent.

**Weaknesses:**

Statistical Reporting Could Be Enhanced: While the results are presented clearly in tables and figures, the paper would be strengthened by more formal statistical testing.
Table 2: The reported performance drops are descriptive (averages). Statistical significance tests (e.g., paired t-tests between baseline and each barrier condition for each metric/model) would solidify the claim that barriers "consistently impair" performance.
Table 3: The comparison between Base, Repair, and (BC+SR) conditions would benefit from statistical tests to confirm that the "modest gains" from BC+SR are statistically significant and that the difference from Repair is meaningful.
Confidence Intervals: It is excellent that confidence intervals are provided for human evaluation (Table 4, Figure 7). It would be beneficial to also include them for the key automatic metrics in the main results (Table 2).
Clarity on Baseline and "Barrier-Free" Performance:
The "barrier-free" baseline is described as one of the four episode sets. It would be helpful to explicitly state its performance in the main results (Table 2) to serve as a clear reference point for the magnitude of degradation.
The text states mutual understanding was "reduced by over 45% on average," but the baseline scores in Table 2 are already quite low for some models (e.g., Mistral's baseline Mutual Understanding is 3.54). A brief discussion on the absolute vs. relative performance drop might be useful.
Minor Writing and Exposition Issues:
The transition between sections can occasionally be abrupt. For instance, the jump from the introduction of the three research questions (Sec. 3) to the first result (Sec. 4.1) could be smoother.
Some acronyms are used before being fully defined (e.g., BC and SR are used in Table 3 before being explicitly expanded in Appendix C.1, though they are described in Sec. 3).

**Questions:**

Generalization and Scalability: The scenarios are adapted from Sotopia. To what extent do you believe the findings and the SOCIALVEIL framework generalize to other interactive benchmarks or entirely new, procedurally generated social scenarios?
Interpretation of Adaptation Results: The results show that interactive learning improves mutual understanding but also increases confusion. What is your interpretation of this trade-off? Does the agent learn to "navigate" the barrier by acknowledging confusion more, without necessarily resolving it?
Barrier Interaction: The study introduces barriers in isolation. In the real world, these barriers often co-occur (e.g., an emotional outburst causing semantic vagueness). How computationally feasible and methodologically sound would it be to study interacting or composite barriers within the SOCIALVEIL framework?
Partner Agent's Role: The framework focuses on the partner agent's resilience. Did you observe any consistent strategies that successful partner agents used to cope with different barrier types? A brief qualitative analysis of such strategies could be a valuable addition.

---

> ### Author Response · Authors · 2025-11-22
> **Official Comment by Authors**
>
> Dear Reviewer wPnq
>
> We'd like to express our sincere gratitude for your thorough review of our paper. We greatly appreciate your suggestions which are crucial in improving the quality of our paper.
>
> **[Enhanced Statisitcal Report]**: We agree that significance tests and confidence intervals can strengthen our claims. Therefore, we performed **paired t-test and computed 95% confidence intervals**. The results confirm that all barrier-induced degradations remain **highly significant** (all p < 0.001), and the confidence intervals of all barrier conditions are non-overlapping with the barrier-free baseline.
>
> Qwen2.5-7B
> | Cond     | GOAL         | BEL          | KNO          | REL          | CONFU        | MUTUAL       |
> | -------- | ------------ | ------------ | ------------ | ------------ | ------------ | ------------ |
> | Base | 7.43±0.26    | 8.56±0.12    | 3.72±0.25    | 3.18±0.21    | 3.89±0.14    | 4.29±0.12    |
> | Sem  | 5.98***±0.23 | 7.48***±0.13 | 2.86***±0.19 | 2.00***±0.14 | 1.75***±0.11 | 1.97***±0.15 |
> | Soc  | 6.01***±0.29 | 7.76***±0.12 | 3.02***±0.17 | 2.31***±0.22 | 2.16***±0.13 | 2.74***±0.17 |
> | Emo  | 5.11***±0.19 | 7.72***±0.10 | 2.71***±0.16 | 1.14***±0.14 | 1.53***±0.08 | 1.79***±0.13 |
>
> Qwen3-4B
> | Cond     | GOAL         | BEL          | KNO          | REL          | CONFU        | MUTUAL       |
> | -------- | ------------ | ------------ | ------------ | ------------ | ------------ | ------------ |
> | Base | 7.73±0.24    | 8.61±0.10    | 3.93±0.24    | 3.18±0.22    | 3.77±0.16    | 4.35±0.11    |
> | Sem  | 6.85***±0.27 | 7.83***±0.10 | 2.89***±0.19 | 1.98***±0.16 | 2.10***±0.13 | 2.43***±0.16 |
> | Soc  | 6.95***±0.24 | 8.06***±0.10 | 3.23***±0.18 | 2.60***±0.15 | 2.51***±0.14 | 3.31***±0.16 |
> | Emo  | 5.90***±0.28 | 7.90***±0.08 | 2.85***±0.14 | 0.95***±0.14 | 1.80***±0.07 | 2.25***±0.11 |
>
> These results show that our created barriers present consistent impaired effect on partner agent performance.
>
> To further evaluate whether adaptation strategies mitigate barrier effects, we also provide the t-test result between Interactive Learning and Base model under barrier conditions:
>
> | Model      | Bar | GOALBase | GOAL_BC | MUT_Base | MUT_BC      | CON_Base | CON_BC      |
> | ---------- | --- | ------ | ----- | ----- | --------- | ----- | --------- |
> | 2.5-7B     | Sem | 5.99   | 6.02  | 1.91  | **2.15*** | 1.61  | 1.84      |
> |            | Soc | 5.71   | 5.87  | 2.45  | **2.60*** | 1.85  | **2.16*** |
> |            | Emo | 5.47   | 5.74  | 2.16  | 2.34      | 1.63  | 1.85      |
> | 3-4B       | Sem | 6.81   | 7.06  | 1.97  | **2.10*** | 2.42  | 2.49      |
> |            | Soc | 6.45   | 6.85  | 2.26  | **2.49*** | 3.02  | **3.21*** |
> |            | Emo | 6.48   | 6.55  | 1.94  | 2.09      | 2.64  | 2.46      |
>
> The test result aligns with our claim that **interactive learning only provides modest gains for mutual understanding, and has minimal effect on social goal completion**
>
> **[Clarity on Baseline and "Barrier-Free" Performance]**: Although the barrier-free condition already appears as the first row in Table 2, we agree that its role as the reference point was not made sufficiently explicit. In the revision, we will highlight the baseline row more clearly and emphasize in the main text that all reported degradations are measured relative to this condition.
>
> Regarding “reduced by over 45%: ”First, both Mutual Understanding and Unresolved Confusion follow a 0–5 Likert scale [1] in which scores rarely approach 5 even for successful interactions. This pattern is aligned with our human evaluation (Table 4): annotators assign a baseline Mutual Understanding of 3.84, closely matching model baselines, indicating that values in the 3.5–4 range should not be interpreted as “low.”
>
> Second, the “barrier-free” setting corresponds exactly to the original SOTOPIA task (identical scenarios, goals, and constraints), meaning the absolute baseline levels reflect the inherent difficulty of SOTOPIA’s open-ended social negotiation tasks rather than any effect introduced by SOCIALVEIL. Prior work on SOTOPIA similarly reports non-saturated scores for both humans and models [2].
>
> Third, the new introduced Mutual Understanding and Unresolved Confusion are validated them through human studies, showing high inter-rater reliability (ICC≈0.78) and strong human–model alignment (r≈0.80), further confirming that the basline rating range is expected. Taken together, these form a consistent and robust triangulation supporting the reasonableness of the baseline values.
>
> [1] Likert, R. (1932). A technique for the measurement of attitudes. Archives of Psychology, 22(140), 1–55.
> [2] Zhou, X. Sotopia: Interactive evaluation for social intelligence in language agents. ICLR 2023
>
> **[Writing Issue]**: Thank you for raising this. We will make sure to fix and improve all the writing issues you mentioned in the updated version!

---

> ### Author Response · Authors · 2025-11-22
> **Official Comment by Authors**
>
> **[Generalization of SocialVeil]**: Thank you for the question. To test whether SOCIALVEIL-trained models gain generalizable social-interaction skills, we evaluate them on AgentSense [1], an external benchmark with scenarios entirely independent of SOTOPIA. Importantly, models never see AgentSense during training, making this a strict zero-shot test.
>
> | Model         | Train Status | Goal Completion |
> |---------------|------------------|----------------|
> | Qwen2.5-7B    | Before Train     | 77.17          |
> | Qwen2.5-7B    | After Train      | 81.31*         |
> | Qwen3-4B      | Before Train     | 82.98          |
> | Qwen3-4B      | After Train      | 86.92*         |
> * indicates statistically significant improvement (p < 0.05).
>
> Both models exhibit clear improvements (+4.14 and +3.94 points, respectively). This indicates that SocialVeil is evaluating the general social intelligence of an intelligent agent, and that training on SocialVeil can provide the model with general skill-level transfer rather than task-specific memorization.
>
> [1] Mou X, et al. AgentSense: Benchmarking Social Intelligence of Language Agents through Interactive Scenarios NAACL 2025
>
>
> **[Regarding adaptation results]** We would like to clarify a possible misunderstanding: In SocialVeil, a higher confusion score indicates less confusion, where we provide details of the Confusion rubrics in Appendix E. Therefore, after adaptation, both Mutual Understanding and Confusion scores increase, indicating that agents get better understanding for each other's goals and also leave fewer confusions unresolved by the end of the interaction.
>
>
> **[Exploration of composite barrier]** We agree that real-world communication often involves multiple barriers simultaneously. In this work, we began with single-factor barriers to ensure controlled causal analysis, allowing us to (i) isolate each barrier’s characteristic failure patterns, and (ii) measure adaptation effects without interactions confounding interpretation.
>
> However, SOCIALVEIL is inherently designed to support composite barriers. Each barrier is implemented as a modular instruction block in the system prompt, so combining barriers (e.g., Semantic + Emotional) requires only concatenating their instruction modules. Our evaluation pipeline also works without modification, making composite-barrier simulation both methodologically simple and computationally inexpensive. To demonstrate feasibility, we constructed a composite Semantic+Emotional barrier and evaluated it using Qwen3-4B. As shown below:
>
> | Cond        | BEL       | GOAL      | KNO       | REL       | MUT       | CONF      |
> | ----------- | --------- | --------- | --------- | --------- | --------- | --------- |
> | Sem     | 7.83±0.10 | 6.85±0.27 | 2.89±0.18 | 1.98±0.16 | 2.43±0.18 | 2.10±0.12 |
> | Emo     | 7.90±0.08 | 5.90±0.28 | 2.85±0.14 | 0.95±0.14 | 2.25±0.12 | 1.80±0.07 |
> | Sem+Emo | 7.36±0.19 | 4.87±0.47 | 2.14±0.24 | 0.93±0.12 | 1.83±0.16 | 1.57±0.11 |
>
> We discover that the composite barrier resulted in the most significant impairment on Goal Completion (4.87), which is notably lower than any single barrier. This confirms the existence of a synergistic negative effect and proves that SOCIALVEIL is uniquely capable of exploring these complex barrier interactions in future work.
>
>
> **[Discussion of partner agent behavior]** We analyze the conversation log before and after interactive learning across three barrier modes and we get following observations:
>
> Semantic: Before adaptation, partner agent often got lost in vague talk. For example, barrier agent turned a price negotiation into a deep discussion about “the meaning of value,” the partner replied in the same abstract way (“Everything has meaning beyond price”).After adaptation, the partner learned to politely acknowledge the comment while bringing the conversation back to the goal: “That’s an interesting thought, but practically—how about $8?”
>
> Sociocultural. Baseline partners tended to echo poetic or sentimental language (“Yes, it holds all our memories”), leading to topic drift. After adaptation, they used gentle clarification questions to re-anchor the dialogue (e.g., “That’s lovely — are you hoping to sell it or keep it as a keepsake?”).
>
> Emotional. Initially, partners relied on rational reassurance (“Every small act matters”), which often intensified distress. After adaptation, they shifted to affective containment — validating emotion and offering low-pressure presence — which improved both relationship (1.2→2.6) and goal completion (5.0→6.8).
>
> Overall, adaptation leads to a generalizable social repair strategy: instead of mirroring the barrier’s style, the partner agent learns to use polite clarifications, grounding moves, and emotional reassurance to guide the conversation back to the shared goal.

---

> ### Author Response · Authors · 2025-11-27
> **Official Comment by Authors**
>
> Dear Reviewer wPnq,
>
> As we approach the end of the rebuttal period, we wanted to kindly follow up. Please let us know if there are any remaining questions or points we can clarify. We truly appreciate your feedback and would be glad to provide any additional information that could support your assessment. If all concerns have been adequately resolved in our responses, we would be grateful if that could be reflected in your scores.
>
> Best regards,
> Authors

---

### Author Response · Authors · 2025-11-23
**Official Comment by Authors**

We appreciate the reviewers’ efforts in evaluating our paper. Below, we summarize the key points reviewers raised—items marked with ** indicate issues for which we provide additional experiments or clarifications, while unmarked items reflect strengths acknowledged by them. "Action/Summary" includes the highly summarized rebuttal content for each reviewer.

| Dimension | Reviewer wPnq | Reviewer NjoN | Reviewer ftGw | Reviewer kqS8 | Action/Summary |
| :--- | :--- | :--- | :--- | :--- | :--- |
| **Novelty & Significance** | "Highly relevant and timely... significant conceptual advance" | "Highlights an important research gap" | "The implementation of different communication barriers is creative.." | "Addresses a clear and important gap" | **Summary:** All reviewers recognize the timeliness and novelty of SocialVeil. |
| **Methodology & Validity** | "Theoretically grounded... Human eval validates fidelity" | "Verifies fidelity through extensive human studies" | **"Unclear if realistic... prompt engineering"** | "Thoroughly validates... t-SNE confirms structure" |  **Summary:** 3 out of 4 reviewers acknowledge methodological validity. **Review ftGw rebuttal:** We clarify SocialVeil is a **principled framework** built on Theoretical Scoping & Methodological Control. Prompting is chosen for interpretability, aligned with standard research. |
| **Evaluation Scope** | "Comprehensive Evaluation... thorough" | "Proposes a comprehensive protocol" | **"Limited... need CoT/Planning"** | "Thoroughly validates its contributions" | **Summary:** 3 out of 4 reviewers believe this paper has a comprehensive evaluation. **Reviewer ftGw rebuttal:** We highlight that **BC+SR (Table 3)** *already* models reflective reasoning (missed by reviewer). Results show barriers persist even with advanced mechanisms. |
|**Evaluation Bias**| NA |**"An ablation with multiple evaluators would strengthen claims.."**|NA|**"...run any experiments where the barrier agent model was varied.."**| **Reviewer NjoN rebuttal:** We emphasize that there is a strong agreement between gpt-4o evaluation and human evaluation on barrier effect, so our claim stands. **Reviewer kqs8 rebuttal:** We add using Qwen2.5-7b as barrier agent to test and show its correlation with using GPT-4o-mini as barrier agent.|
| **Generalization** | **"To what extent do you believe the findings and the SOCIALVEIL framework generalize to.."** | NA | **"It is unclear whether these prompts truly capture realistic sociocultural..."** | NA | **Reviewer wPnq & ftGw rebuttal:** We conducted additional experiments evaluating SocialVeil-trained agents on a separate social learning benchmark to verify generalization. |
| **Presentation** | "Commendable commitment to reproducibility" | "Nicely presented and easy to follow" | "Clearly written and organized" | "Well-written with clear figures" | **Summary:** All reviewers agree the paper is clearly presented and well-structured. |

In addition, we expand several minor discussions: such as the qualitative case study of strategies learned via interactive learning, the feasibility of studying composite barriers, as well as the specific mechanisms driving sociocultural confusion. These clarifications do not affect the core contribution of the paper and are fully addressed in the rebuttal.

---

### Meta-Review · Area_Chair_imV9 · 2026-01-07

**Summary:**

The reviewers agreed the paper is well-motivated and SOCIALVEIL is clearly presented.
The suggested decision was primarily informed by concerns about
- whether the contribution is a principled, validated benchmark versus largely prompt engineering
- robustness and bias from fixing the barrier agent and the evaluator to a single LLM
- generalization beyond Sotopia-derived scenarios
- experimental rigor/breadth, including formal significance testing, clearer baselines, and evaluation on stronger frontier models and/or more agentic reasoning setups.

**Reviewer Concerns:**

(Partially) addressed by the rebuttal:
- Statistical testing & confidence intervals (wPnq): Authors report paired t-tests and 95% CIs for barrier-induced degradations.
- Generalization beyond Sotopia (wPnq, NjoN): Authors evaluate on an external benchmark, AgentSense, as a strict zero-shot test.
- Evaluator bias from single LLM judge (NjoN): Authors argue main conclusions are corroborated by human evaluation; they report strong human–LLM agreement and correlation, reducing concern that conclusions are an artifact of GPT-4o judging.
- Frontier-model coverage (NjoN): Authors add results for GPT-5.
- Single barrier-agent (kqS8): Authors vary the barrier agent (e.g., Qwen2.5-7B).
- Composite barrier feasibility (wPnq): Authors add a composite Semantic+Emotional condition.
- Partner-agent behavior / qualitative strategies (wPnq): Authors include qualitative observations (e.g., grounding moves, polite clarifications, affective containment) describing how adaptation changes partner strategies.

Still outstanding:
- limited novelty (ftGw, kqS8): reviewer's main concerns are SocialVeil might be prompt engineering rather than methodological advance. Even with theory grounding and validation, reviewers' core critique remains that barrier injection is essentially prompt templating and that results are expected; rebuttal reframes as a validated diagnostic framework but may not convince reviewers.
- Meaningfulness (ftGw, NjoN): Concerns that prompts may not capture real sociocultural/emotional phenomena.
- Limited evaluation breadth with more agentic/reasoning systems (ftGw, wPnq): While authors discuss repair prompting and interactive learning, requests for broader agent architectures (reflection/planning/coordination, stronger reasoning scaffolds) is not fully satisfied.
- Code availability at review time (NjoN): Authors promise code/data release at camera-ready; reproducibility remains contingent until release.

**Reviewer Scores:**

wPnq (4): Likely added to 6 given added statistical rigor, clearer baseline framing.
NjoN (4): likely unchanged or added to 6 after added frontier-model results, external-benchmark generalization, and mitigation of evaluator-bias concerns via human evaluation.
ftGw (2): unlikely to change the score. Core novelty concerns remain.
kqS8 (6): likely unchanged.

---

### Decision · Program_Chairs · 2026-01-26

Reject